# FineGRAIN: Evaluating Failure Modes of Text-to-Image Models with Vision Language Model Judges

**Kevin David Hayes**
University of Maryland
khayes1@umd.edu

**Micah Goldblum**
Columbia University
micah.g@columbia.edu

**Gowthami Somepalli**
University of Maryland
gowthami@umd.edu

**Vikash Sehwag**
Sony AI*
sehwag.vikash@gmail.com

**Ashwinee Panda**
University of Maryland
ashwinee@umd.edu

**Tom Goldstein**
University of Maryland
tomg@umd.edu

https://finegrainbench.ai

## Abstract

Text-to-image (T2I) models are capable of generating visually impressive images, yet they often fail to accurately capture specific attributes in user prompts, such as the correct number of objects with the specified colors. The diversity of such errors underscores the need for a hierarchical evaluation framework that can compare prompt adherence abilities of different image generation models. Simultaneously, benchmarks of vision language models (VLMs) have not kept pace with the complexity of scenes that VLMs are used to annotate. In this work, we propose a structured methodology for jointly evaluating T2I models and VLMs by testing whether VLMs can identify 27 specific failure modes in the images generated by T2I models conditioned on challenging prompts. Our second contribution is a dataset of prompts and images generated by 5 T2I models (Flux, SD3-Medium, SD3-Large, SD3.5-Medium, SD3.5-Large) and the corresponding annotations from VLMs (Molmo, InternVL3, Pixtral) annotated by an LLM (Llama3) to test whether VLMs correctly identify the failure mode in a generated image. By analyzing failure modes on a curated set of prompts, we reveal systematic errors in attribute fidelity and object representation. Our findings suggest that current metrics are insufficient to capture these nuanced errors, highlighting the importance of targeted benchmarks for advancing generative model reliability and interpretability.

## 1 Introduction

Vision-Language Models (VLMs) have become essential tools in multimodal AI, enabling systems to interpret and answer questions about images and text. Despite these advancements, VLMs still lack key capabilities, particularly in compositional reasoning. Studies such as Huang et al. [14] highlight that VLMs struggle to handle complex scenes that involve multiple objects, attributes, or interactions. To address these limitations, researchers have developed a multitude of benchmarks aimed at identifying VLM failure modes. However, the variety of these benchmarks and their often narrow focus make it challenging for developers to select an evaluation that aligns with their application needs, even with benchmark aggregations like Al-Tahan et al. [2]. This benchmarking

---

*Now at Google Deepmind

39th Conference on Neural Information Processing Systems (NeurIPS 2025) Track on Datasets and Benchmarks.

Figure 1: Overview of FineGRAIN architecture

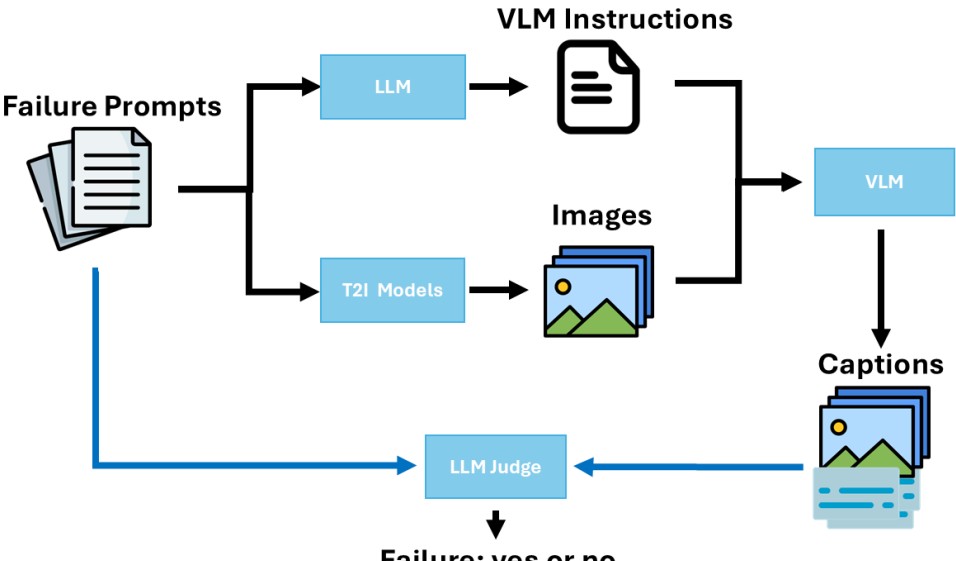

gap underscores the need for a structured evaluation framework that can help developers diagnose specific limitations in VLMs relevant to their goals.

In parallel, text-to-image (T2I) models, especially diffusion-based models, are transforming creative and generative AI applications but continue to face challenges with prompt adherence. Although T2I models are widely used in open-source projects, reliability issues have constrained their broader adoption, particularly in commercial applications where prompt fidelity is critical. Like VLMs, T2I models struggle with generating outputs that satisfy specific requirements, such as correct object counts or color bindings. These challenges reveal interlinked issues of model reliability and capability, highlighting the need for a structured approach to understanding where these models succeed or fail in responding accurately to user prompts.

In this work, we present FineGRAIN: a joint evaluation framework to rigorously assess both VLMs and T2I models through a structured, category-specific lens. Our approach pairs diffusion models with VLMs and evaluates them using a curated set of prompts designed to trigger specific errors in prompt adherence. We define 27 distinct failure modes, such as object miscounts and incorrect attribute bindings, and generate 25–30 prompts aimed at eliciting each of these failure modes. For the evaluation, we create a dataset of over 3,750, 1360x768 resolution images generated by five T2I models (Flux, SD3-Medium, SD3-Large, SD3.5-Medium, SD3.5-Large) and annotated by a VLM (Molmo, InternVL3, Pixtral) and LLM (Llama3). This framework grades the VLMs on whether they correctly identify discrepancies between the prompts and generated images, offering an in-depth view of model performance across failure categories.

Our contributions include a curated dataset, a structured methodology for evaluating prompt adherence, and a flexible evaluation tool for developers. By enabling application-specific failure mode analysis, our framework provides insights into the unique weaknesses of both VLMs and T2I models, helping to advance the development of more reliable, interpretable multimodal AI systems. This approach not only addresses current evaluation gaps but also offers a durable resource for benchmarking future models across varied and nuanced prompt categories.

## 2   Related Work

### 2.1   Text-to-Image (T2I) Generative Models

In this work, we primarily evaluate the capabilities of open- source diffusion-based generative models, such as Stable-diffusion [9, 26], and Flux [5], that generate images conditioned on text prompts.

While the design of conditioned image generative models can support a wide range of conditioning signals (conditioned on other modalities like audio [4]), text-to-image (T2I) generation is the most widely explored. In large-scale T2I models, the goal is to enable generalization to a diverse range of prompts, varying in length and complexity, while also providing strong alignment between the prompt and the generated image [16, 17, 20, 34]. As it is challenging to obtain a large dataset of high-quality image-caption pairs in the real world, T2I models are often trained on detailed synthetic captions generated using image-captioning models [3]. Most large-scale T2I models now commonly employ diffusion transformer architectures [24] and are trained on billions of images [28]. In addition to text captions, multiple T2I models also support additional conditioning on image resolution to enable image generation at varying resolution [25].

## 2.2 T2I and VLM Evaluation

Li et al. [21] propose a new dataset with 1600 prompts focusing on compositional reasoning for T2I models, and uses VQAScore [22] to rank different images. VQAScore is a metric that takes in an image and a text and outputs the likelihood that the image contains the text. They find that VQAScore outperforms other commonly used metrics, such as CLIPScore [11] and PickScore [18]. However, VQAScores are high in general, and as we will show, they have difficulty identifying tasks where models have high failure rates.

Fu et al. [10] propose an instruction following benchmark for T2I models that focuses on what they call "adversarial" prompts. Unfortunately, these prompts fail to capture the complexity of real use cases, as half of the 150 prompts contain fewer than 5 words. Many of their prompts are not long enough to even have a definitive outcome. For example: "A sundae left alone for several hours" is a prompt they expect to generate melted ice cream, but the prompt does not specify that it's outside on a hot day. We create an entire failure mode for Negation, and our negations are much more diverse, thorough, and specific (full list of all negation prompts deferred to Appendix).

Shahgir et al. [29] propose a VLM benchmark based on challenging prompts, such as optical illusions. We also evaluate VLMs on their capability to answer questions about nonsensical images, via the "Opposite of Normal Relation" and "Negation" failure modes. Our dataset includes these modes, and also 25 other failure modes and additionally evaluates T2I models. By performing these evaluations in a unified framework, we are able to answer the question not only of "How well can T2I/VLMs do X", but also *relatively* how well T2I/VLMs can do X as compared to other tasks.

TIFA Hu et al. [12], DSG [7] and Gecko Wiles et al. [32], propose using automatically generated questions for generated images. These questions lack state-of-the-art failure mode tailored questions, instead leading to evaluations around already solved capabilities while lacking adequate testing of the skills that adequately differentiate the state-of-the-art-models. Gecko, for example, uses automatic tagging.

One commonality of prior benchmarks is that they adopt a coarse view of T2I/I2T capabilities. As we will show, fine-grained decompositions of broad concepts are import to identify the key deficiencies in T2I and VLM capabilities. One motivating example: Li et al. [21] observe that SDXL does well on counting. However, we will show the exact opposite conclusion, because we go into detail on counting prompts and observe in Table 7 that SDXL's performance drops off sharply when asked to generate more than a very small number of things.

**Gaps in the State of the Art.** Across all prior work, we see that all existing benchmarks evaluate either T2I models or VLMs. Furthermore, benchmarks tend to focus on niche facets of the failure landscape. FineGRAIN is a more comprehensive vision benchmark because we aim to evaluate both T2I and VLM across all failure categories on the same benchmark. Furthermore, by evaluating both pieces of the pipeline, we critique our own evaluation methods making them more reliable. This helps better understand the opportunities of T2I reward modeling and mitigate its challenges like reward hacking for a given failure mode.

## 3 The FineGRAIN Framework

FineGRAIN is our framework for **G**enerating **R**atings with **A**gents for **I**magi**N**g. In this section we outline the design of our FineGRAIN framework and our methodology for creating our dataset.

Table 1: The LLM instruction prompt and one example of the failure mode-specific template. All templates can be found in the Appendix.

```
Create an instruction prompt for the diffusion ''prompt''.  Create the instruction
prompt by using the templates below based on which failure mode the diffusion prompt
is.
Use the prompt and nothing else.  The only thing that should change in the
instruction prompt is to replace anything that is in brackets [ ] with those
categories from the diffusion prompt.  If prompt is in the brackets input the entire
prompt in quotes.
If there are more or less than the required brackets they can be added or removed,
though not typical.  There are additionally instructions or advice specific to each
failure mode in quotes or labeled as guidance below the template for you to make the
best prompt from the template.  Do not output anything other than the instruction
prompt tailored to the diffusion prompt.
```
```
"Counts or Multiple Objects":  { "template":  "Count how many [object] are there?
Count how many [object] are there?  Count how many [object] are there?", "guidance":
"Objects will be numbered more than one" }
```

Table 2: High-Level Categorization of Failure Modes. Some specific failure modes are present in multiple high-level categories. Table 3 shows how we have covered new high-level categories as compared to prior work. Note that prior work does not have finegrained categories, only coarse high-level categories.

| High-Level Category | Failure Modes |
|---|---|
| Scene | Background and Foreground Mismatch, FG-BG relations |
| Attribute | Color attribute binding, Shape attribute binding, Texture attribute binding, Counts or Multiple Objects, Scaling, Perspective |
| Relation | Spatial Relation, Physics, FG-BG relations, Background and Foreground Mismatch, Visual Reasoning Cause-and-effect Relations |
| Count | Counts or Multiple Objects, Social Relations |
| Negation | Negation |
| Differ | Scaling, Social Relations, Surreal, Depicting abstract concepts, Emotional conveyance, Social Relations, Human Action |
| Human | Action and motion representation, Anatomical limb and torso accuracy, Emotional Conveyance, Human Action, Human Anatomy Moving, Social Relations |
| Text | Text-based, Short Text Specific, Long Text Specific, Tense+Text Rendering + Style |
| Multi-Style | Blending Different Styles, Opposite of Normal Relation, Surreal, Background and Foreground Mismatch, FG-BG relations, Texture attribute binding |
| Adversarial | Opposite of Normal Relation, Surreal, Background and Foreground Mismatch, Depicting abstract concepts |
| Temporal | Human Action, Visual Reasoning Cause-and-effect Relations, Tense and aspect variation, Tense+Text Rendering |

## 3.1 The FineGRAIN Agents.

FineGRAIN is an agentic system for rating Text-to-Image and Image-to-Text models by determining whether a VLM can identify anything wrong with an image generated by a T2I model. If the T2I instruction is "Three dogs", we ask the VLM "How many dogs are in this image?" and if it's not three, the T2I model has failed to follow our instructions. An LLM automatically creates the "How many dogs are in this image?" prompt for the VLM given the T2I instruction and our manually created instruction prompt. The instruction prompt for the LLM is conditioned on the failure category as shown in Table 1. The LLM also compares the VLM's answer to the T2I instruction to determine whether the T2I model has failed. In this manner, we can grade the T2I model. In order to grade the VLM, or VLM+LLM, we need to compare the automated pipeline answer to the human ground truth. Ultimately, the output of the FineGRAIN pipeline is a boolean score indicating whether the image complies with the user prompt, a raw score that can be used for ranking images, and an explanation for the score.

Table 3: **Comparing FineGRAIN to existing text-to-visual benchmarks.** FineGRAIN covers a broader range of skills than prior benchmarks, even at a coarse granularity. Table 2 shows the categorization of our finegrained failure modes into these coarsely organized skills.

| Benchmarks | Skills Covered in Prior Benchmarks | | | | | | Additional FineGRAIN Categories | | | | |
| --- | --- | --- | --- | --- | --- | --- | --- | --- | --- | --- | --- |
| | Scene | Attribute | Relation | Count | Negation | Differ | Human | Text | Multi-Style | Adversarial | Temporal |
| PartiPrompt (P2) [35] | ✓ | ✓ | ✓ | ✓ | ✓ | ✗ | ✗ | ✗ | ✗ | ✗ | ✗ |
| DrawBench [27] | ✓ | ✓ | ✓ | ✓ | ✗ | ✗ | ✗ | ✗ | ✗ | ✗ | ✗ |
| EditBench [31] | ✓ | ✓ | ✓ | ✓ | ✗ | ✗ | ✗ | ✗ | ✗ | ✗ | ✗ |
| TIFAv1 [13] | ✓ | ✓ | ✓ | ✓ | ✗ | ✗ | ✗ | ✗ | ✗ | ✗ | ✗ |
| Pick-a-pic [19] | ✓ | ✓ | ✓ | ✓ | ✗ | ✗ | ✗ | ✗ | ✗ | ✗ | ✗ |
| T2I-CompBench++ [15, 17] | ✓ | ✓ | ✓ | ✓ | ✗ | ✗ | ✗ | ✗ | ✗ | ✗ | ✗ |
| HPDv2 [33] | ✓ | ✓ | ✓ | ✗ | ✗ | ✗ | ✗ | ✗ | ✗ | ✗ | ✗ |
| EvalCrafter [23] | ✓ | ✓ | ✓ | ✓ | ✗ | ✗ | ✗ | ✗ | ✗ | ✗ | ✗ |
| GenAIBench [21] | ✓ | ✓ | ✓ | ✓ | ✓ | ✓ | ✗ | ✗ | ✗ | ✗ | ✗ |
| **FineGRAIN (Ours)** | ✓ | ✓ | ✓ | ✓ | ✓ | ✓ | ✓ | ✓ | ✓ | ✓ | ✓ |

**An example use of FineGRAIN.** We now provide an example of the boolean score and reasoning. The LLM judge's output is the following: "The failure mode is present because the model has inaccurately rendered the long specific text on the welcome sign. The original prompt has 'Each stone, each artifact tells a story of a time long past', but the caption has 'Each story, art, and artifact tells a tale of our past'. The model has changed the wording and added 'story' and 'art' which affects the legibility and integration with other elements." The corresponding boolean score is 1, indicating that the failure mode (in this case, long text generation) is present. The raw score that we can use for ranking images is 8.0.

## 3.2 New Capabilities of FineGRAIN.

**Boolean score.** The first new capability of FineGRAIN is that we get a boolean score; "did the T2I fail to follow the user's instructions", whereas prior scores such as VQAScore and CLIPScore are not designed with this capability in mind. Li et al. [21] apply VQAScore primarily to rank different images, and we can also use FineGRAIN for this. However, the appeal of a boolean score is that we can deploy a T2I model into a pipeline where we continuously generate images until we generate an image that FineGRAIN determines to have complied with the user's instructions. This is an element of test-time scaling that we contend will be especially valuable for T2I deployments.

**Objective Human Annotations.** Human ratings that take into account aesthetics are inherently subjective. We design FineGRAIN to primarily focus on prompts where the human rating can be seen as objective. For counting or text rendering, the human score is entirely unambiguous, and here FineGRAIN has a high correlation with the human label.

**Interpretable Scores.** Prior work does not offer interpretable scores, while our agentic workflow produces the LLM judge's reasoning for determining whether the image is compliant with user instructions. The interpretability of FineGRAIN is a major asset for diagnosing why models fail to comply with user instructions.

## 3.3 An Ontology of Failure Modes

We identify 11 high-level kinds of failure modes in T2I generation by analyzing user complaints. We further split up these high-level failure categories, resulting in 27 specific *fine-grained* failure modes as shown in Table 2. We then hand-write 25-30 prompts for each failure mode to elicit different kinds of failures, with examples given in the Appendix. This high level of human curation makes our coverage of failure modes more comprehensive and importantly more *fine-grained* than prior work, as we outline in Table 3.

## 4 Analysis and Results

**Models.** We use open source models for each component in our pipeline. The LLM is Llama3-70B [30]. The chosen VLM is Molmo-72B [8]; while also using InternVL-78B [6] and Pixtral-124B [1] during VLM testing. In the Appendix we provide a comparison between different VLMs. We evaluate 5 T2I models: Flux-dev [5], Stable Diffusion XL [25], Stable Diffusion 3 Medium [9],

Stable Diffusion 3.5, and Stable Diffusion 3.5 Medium. In this work we focused on open-source models. We are continuously evaluating more open-source and closed-source models adding to our benchmark website (https://finegrainbench.ai/) and Huggingface Repo.

**Human Data Annotation.** Every prompt is tagged with exactly 1 finegrained failure mode. Each prompt has 5 outputs images for each T2I models, and each image is annotated by a human with a ground-truth label. This label is 1 if the image contains a failure mode, and 0 otherwise.

## 4.1 T2I Evaluation

In Table 4 we give examples of all 5 model outputs on single prompts sampled from a subset of failure modes. All prompts, the T2I generations, and the human ground-truth can be found in the Appendix.

Table 4: Samples of 5 models on 4 failure modes.

| Flux | SD35 | SD35-M | SD3-XL | SD3-M |
| --- | --- | --- | --- | --- |

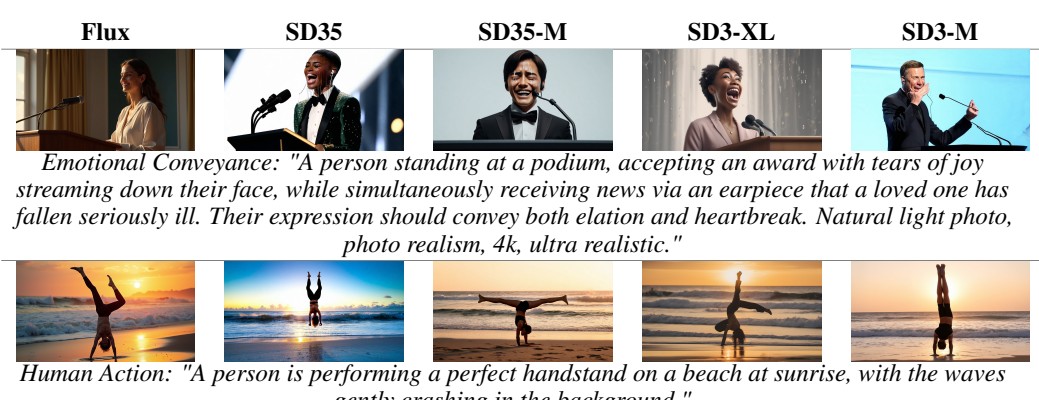

*Emotional Conveyance: "A person standing at a podium, accepting an award with tears of joy streaming down their face, while simultaneously receiving news via an earpiece that a loved one has fallen seriously ill. Their expression should convey both elation and heartbreak. Natural light photo, photo realism, 4k, ultra realistic."*

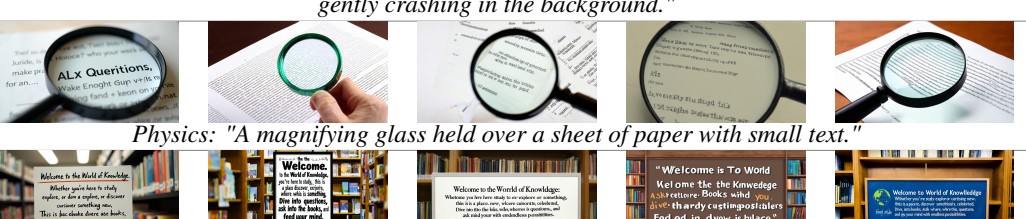

*Human Action: "A person is performing a perfect handstand on a beach at sunrise, with the waves gently crashing in the background."*

*Physics: "A magnifying glass held over a sheet of paper with small text."*

*Text: "A motivational sign in a library reading: Welcome to the World of Knowledge. Whether you're here to study, explore, or discover something new, this is a place where curiosity is celebrated.¨*

In Table 5 we compare the performance of 5 different T2I models across all 27 failure modes. The score is how often the model succeeds in generating the image correctly given the prompt, as judged by humans. We immediately see the benefits of taking a finegrained approach to T2I evaluation by seeing that some categories that were simply lumped in with others actually have very different success rates. For example, all models fail completely to generate "Counts or Multiple Objects" so we know that these models struggle to generate the correct numbers of objects. In prior work such as Li et al. [21], counting is mixed together with other potential failure modes. For example, "Six oval stones" could receive a poor score from the human because it had the wrong number of stones, or because the stones were not oval. Our evaluation separates these two failure modes.

### 4.1.1 Prompt Difficulty Ablations

A common complaint against new benchmarks is that models eventually saturate all evaluations. We argue that FineGRAIN offers a unique opportunity to adjust the difficulty of the evaluation. We focus on two failure modes: generating long text and counting multiple objects. We show that the difficulty of the prompts can be adjusted near-programmatically, and that even the best-performing models still have a lot of room for improvement.

**Generating text.** We include multiple distinct failure modes for text; Text Rendering Style, Text-Based, Short Text Specific, and Long Text Specific. The results in Table 6 show a clear decrease

Table 5: Model Diffusion Performance as graded by a binary Human Evaluation of each failure mode

| Failure Mode | Flux | SD3.5 | SD3.5 M | SD3 M | SD3 XL |
|---|---|---|---|---|---|
| Cause-and-effect Relations | **44.83** | 36.84 | 31.58 | 27.59 | 21.05 |
| Action and Motion | **52.00** | 20.00 | 16.00 | 0.00 | 12.00 |
| Anatomical Accuracy | **53.33** | 33.33 | 26.67 | 6.67 | 26.67 |
| BG-FG Mismatch | **76.00** | 69.23 | 73.08 | 53.85 | 53.85 |
| Blending Styles | 5.00 | 10.34 | 3.45 | **13.79** | 3.45 |
| Color Binding | 93.33 | **96.67** | 93.33 | **96.67** | 40.00 |
| Counts or Multiple Objects | 0.00 | 0.00 | 0.00 | 0.00 | 0.00 |
| Abstract Concepts | **92.31** | 84.62 | 88.46 | 73.08 | 69.23 |
| Emotional Conveyance | **76.67** | 46.67 | 36.67 | 16.67 | 33.33 |
| FG-BG Relations | **86.21** | 37.93 | 34.48 | 51.72 | 37.93 |
| Human Action | **72.41** | 68.97 | 27.59 | 13.79 | 44.83 |
| Human Anatomy Moving | **79.31** | 48.28 | 17.24 | 0.00 | 24.14 |
| Long Text Specific | 0.00 | 0.00 | 0.00 | 0.00 | 0.00 |
| Negation | 25.00 | **46.43** | 46.43 | 17.86 | 46.43 |
| Opposite Relation | **6.67** | **6.67** | 3.33 | 0.00 | 0.00 |
| Perspective | **33.33** | 23.33 | 20.00 | 10.00 | 6.67 |
| Physics | **43.33** | 16.67 | 23.33 | 26.67 | 16.67 |
| Scaling | **43.33** | 33.33 | 26.67 | 23.33 | 23.33 |
| Shape Binding | **60.00** | 50.00 | 30.00 | 30.00 | 3.33 |
| Short Text Specific | **64.00** | 48.00 | 24.00 | 20.00 | 0.00 |
| Social Relations | **84.62** | 65.38 | 30.77 | 7.69 | 34.62 |
| Spatial Relations | **50.00** | 23.33 | 16.67 | 23.33 | 10.00 |
| Surreal | 28.00 | **44.00** | 36.00 | 36.00 | 12.00 |
| Tense and Aspect | **57.69** | 42.31 | 38.46 | 42.31 | 23.08 |
| Text Rendering Style | **28.00** | 4.00 | 0.00 | 0.00 | 0.00 |
| Text-Based | **79.31** | 62.07 | 27.59 | 27.59 | 3.45 |
| Texture Binding | 43.33 | **63.33** | 53.33 | 36.67 | 23.33 |
| **Average** | **51.04±1.83** | 40.06±1.79 | 30.56±1.68 | 24.27±1.57 | 21.09±1.49 |

in text generation success as token count increases, with success rates dropping from 0.520 for three-token prompts to 0.136 for ten-token prompts, and reaching 0.0 by fifty tokens. Among individual models, SD 3.5 Large achieves the highest success rate for short, three-token prompts at 0.92, outperforming others by a notable margin; however, its performance drops to 0.28 for ten tokens and reaches zero for longer sequences.

Table 6: Comparison of performance of Flux, SD3-Medium (3-M), SDXL, SD3.5-Large (3.5-L), SD3.5-Medium (3.5-M) in generating correct text in images. While Flux and SD3.5-Large are quite good at generating short phrases, the rate of success sharply decays as text quantity (3 tokens, 10 tokens, 20 tokens, 50 tokens) increases.

| Model | 3 Tokens | 10 | 20 | 50 | Avg |
|---|---|---|---|---|---|
| Flux | 0.84 | **0.40** | **0.04** | 0.00 | **0.32$_{0.4}$** |
| 3-M | 0.44 | 0.00 | 0.00 | 0.00 | 0.11$_{0.3}$ |
| 3.5-L | **0.92** | 0.28 | 0.00 | 0.00 | 0.30$_{0.4}$ |
| 3.5-M | 0.40 | 0.00 | 0.00 | 0.00 | 0.10$_{0.3}$ |
| SDXL | 0.00 | 0.00 | 0.00 | 0.00 | 0.00$_{0.0}$ |
| **Avg** | **0.520** | **0.136** | **0.008** | **0.00** | |

Table 7: Comparison of performance of Flux, SD3-Medium (3-M), SDXL, SD3.5-Large (3.5-L), SD3.5-Medium (3.5-M) in generating correct numbers of objects in images. All models can generate a single object with some consistency, but performance quickly degrades as the number of objects increases.

| Model | 1 Obj. | 2 Obj. | 3 Obj. | Model Avg. |
|---|---|---|---|---|
| Flux | **0.655** | **0.103** | 0.034 | **0.264$_{0.1}$** |
| SD 3 Medium | 0.379 | 0.034 | 0.034 | 0.149$_{0.1}$ |
| SD 3.5 Large | 0.483 | 0.069 | **0.103** | 0.218$_{0.1}$ |
| SD 3.5 Medium | 0.345 | 0.034 | 0.034 | 0.138$_{0.1}$ |
| SDXL | 0.138 | 0.034 | 0.034 | 0.069$_{0.1}$ |

**Counting.** The results in Table 7 indicate a pronounced difficulty among models in handling prompts with multiple objects, as success rates consistently decline with increased object counts. For single-object prompts, Flux achieves the highest success rate at 0.655, followed by SD 3.5 Large at 0.483. However, when tasked with two or three objects, all models experience substantial drops, with Flux maintaining only 0.103 success for two-object prompts and 0.034 for three objects.

Table 8: Prompt: "A person hitting a hard drum that has sand on the drum". Only FineGRAIN can find the correct image (SD3.5). FineGRAIN outputs 1 if the image contains a failure mode and 0 otherwise.

| | FLUX | SD3-M | SD35-M | SD35 | SD3-XL |
|---|---|---|---|---|---|
| **Images** | | | | | |
| **fineGRAIN (LLM Boolean)** | 1 | 1 | 1 | 0 | 1 |
| **VQAScore** | 0.893 | 0.967 | 0.717 | 0.909 | 0.954 |
| **CLIP Score** | 0.316 | 0.273 | 0.314 | 0.266 | 0.328 |

## 4.2 VLM Evaluation

We have established how we evaluate different T2I models. We can now move towards evaluating the VLM by determining whether its captions accurately capture the failure modes as annotated by the human. In this section, we primarily evaluate the VLM+LLM together because we find that the results are not significantly different from evaluating the VLM individually. We defer an evaluation of the VLM itself to the Appendix. Throughout this subsection, we compare the VLM to the prior SOTA, VQAScore.

VLMs struggle with many of the failure modes that text to image diffusion models do as well, suggesting that these are problems with the vision itself. That said, VLMs are still useful as reward models, as their failure rates are generally lower than those of text-to-image models for the same failure modes. This is likely in part due to the difference in information richness and density comparing image and text modalities. For example, our best text-to-image model code does not reliably generate the correct number of multiple objects however our best VLMs have a decent success rate at picking up these objects numbers. Another advantage VLM models have over text-to-image modes is that they have multimodal guidance. We optimize this multimodal guidance by tailoring the text modality of the instruction prompt to each specific image.

**VLMs Cannot Be Trusted.** One finding in our work that sharply differs from prior work is that we give all models a very low score for counting, under the failure mode "Counts or Multiple Objects". This is primarily because prior work has been very lax at assessing whether the model is actually generating the exact right number of objects.GenAIBench [21] uses VQAScore [22], which gives the VLM the prompt directly. Therefore, they rely on the VLM to correctly determine whether "3 bananas" is in the image. In the appendix we ablate how the performance of a VLM changes if we actually give the VLM the prompt that we are asking it to evaluate, as Lin et al. [22] do.

### 4.2.1 Comparing FineGRAIN to VQAScore

VLMs still struggle with understanding images that are outside of their training data and are biased towards out-of-distribution data. This can be compounded by instruction prompts that lead the model to certain conclusions. We observed a drop in model performance for certain failure modes when the original text-to-image prompt was also shown to the VLM model, the VLM model being more likely to confirm the accuracy of the prompt. Distortions to human anatomy are often overlooked by VLM models and when seen, are even explained by the model as an optical illusion.

In Table 8 we show an example of a prompt where VQAScore and CLIPScore are unable to identify the correct image. For the prompt "A person hitting a hard drum that has sand on the drum", the correct image should show sand flying up off the surface of the drum as it is hit. Only FineGRAIN correctly identifies the correct image (generated by SD3.5).

**Failure Detection.** In our dataset, each prompt-image pair has a human ground-truth boolean label of "did this image comply with the user instruction", for 5 different T2I models. We first filter for challenging-but-doable prompts, where at least one model fails and at least one model succeeds.

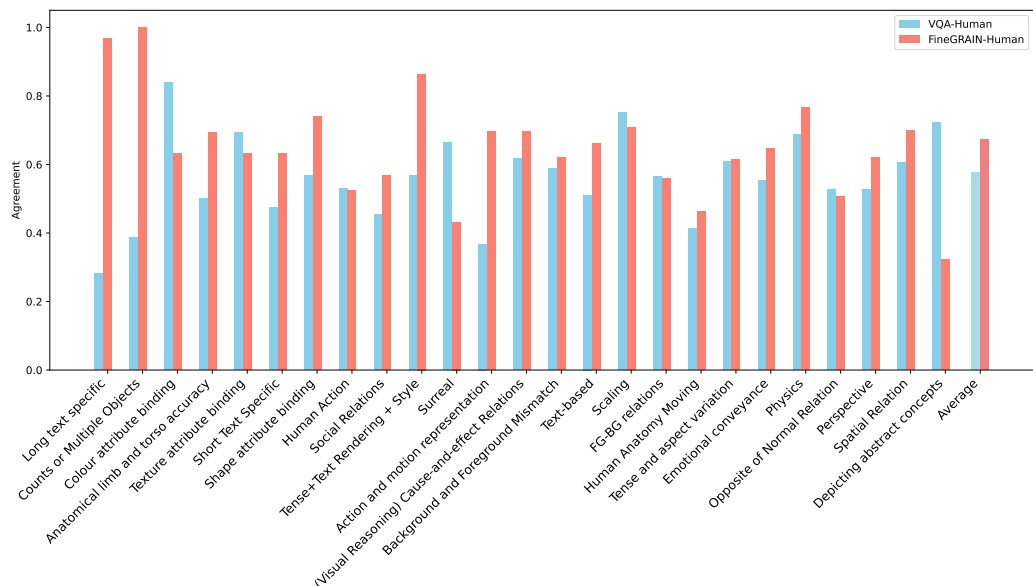

Figure 2: Comparison of agreement rates with human ground truth between VQAScore and Fine-GRAIN. FineGRAIN outputs a boolean prediction of whether the image contains a failure mode. VQAScore is a numerical score that we threshold to obtain a boolean (we ablate this threshold in the Appendix). FineGRAIN generally outperforms VQAScore.

We first evaluate how well FineGRAIN can determine whether the image complies with the user instruction as compared to the SOTA metric VQAScore. In Figure 2 we plot the agreement rate with the human ground truth boolean label of whether the image complies with the user instruction, for both VQAScore and the FineGRAIN boolean prediction. We convert VQAScore to a boolean by thresholding it at 0.9. We provide a full ROC curve for VQAScore in the Appendix.

**VQAScore-Human Agreement.** We find that the average VQAScore-Human agreement is 57.7%. VQAScore is a particularly poor judge on both short and long text, where it agrees with the human ranking < 30% of the time. The category where VQAScore has the highest accuracy is color attribute binding, where it achieves 84%.

**FineGRAIN-Human Agreement.** The average FineGRAIN-Human agreement is 67.4%, a 10% improvement over the VQAScore-Human agreement. While FineGRAIN achieves near-human performance for some categories such as "Counts or Multiple Objects" and "Long text specific", it performs quite poorly for others. For example, it diverges from the human rating on more than 50% of prompt-image pairs in the "Surreal" failure mode. Arguably, categories such as "Surreal" are not as *objective* as the rest of our evaluation. In general, we find that the FineGRAIN failure prediction is well aligned with the human label on Flux.

## 5 Discussion

In this work, we primarily focus on evaluating VLM and T2I models on criteria that are failure modes in their generation or understanding. This is a departure from prior work, that has mostly sought to rank T2I models according to their aesthetic abilities or general questions. We choose to mostly target objective criteria because we feel that as T2I models become increasingly capable, they are no longer differentiated by whether they generate prettier images, but whether they do not display failure modes. It is easier to source ground-truth human annotations for images when the annotation just needs to be a binary number indicating whether the image contains the correct number of bananas, than to ask multiple raters to grade images according to subjective criteria.

**Limitations.** In the main paper, we only consider a single LLM (Llama3-70B) in our FineGRAIN pipeline. Other LLMs and especially closed-source models may perform better. In the same vein, we made the decision to use only open-source VLMs in our evaluation, despite closed-source models

performing slightly better on most tasks. The best performing LLMs and VLMs are quite large which can make the optimal approach expensive. We provide a comparison between these VLMs in the Appendix. Our VLM and LLM judges also have failure modes that hamper their ability to evaluate diffusion models.

## Broader Impact

This paper's aim is to advance text-to-image and image-to-text modeling. Our work could be used to advance the evaluation, understanding and accuracy of these models. Thus, any general societal impact of these models' successes or faults could come under the impact of this paper when our evaluation or dataset is used to further it. Text-to-image models have many positive impacts in allowing the quick rendering of images for a number of fields. That said, if they become too accurate there could be negative impacts if it becomes difficult to tell real images from fake images.

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

# A Appendix

Table 9: Sample Prompts for Each Failure Mode

| Failure Mode | Description | Sample Prompt |
|---|---|---|
| Counts or Multiple Objects | The model struggles with generating a precise number of distinct objects in a scene. | An arrangement of exactly two red apples and precisely three yellow bananas on a circular plate. Blur background, product photography. |
| Color attribute binding | The model has difficulty correctly associating colors with specific objects in a scene. | A miniature red sheep driving a white car, Pixar-style 3D rendering, highly detailed. |
| Shape attribute binding | The model confuses or incorrectly generates shapes for objects. | A surreal landscape featuring a large, pyramid-shaped cloud floating in the sky. Below it, a circular lake reflects the cloud and sky. The scene has soft, pastel colors. Hyperrealistic rendering, 8k resolution. |
| Texture attribute binding | The model incorrectly applies textures to objects. | An award-winning photo of a cute marble boat with visible veining, floating on a rough sea made entirely of sandpaper. |
| Spatial Relation | The model struggles with accurately placing objects in relation to each other. | A puppy balanced precariously on the head of a patient dog, studio lighting, high detail, 4K resolution. |
| Physics | The model fails to generate or follow the innate physical laws in the scene. | A ball with a very low elastic modulus hitting a solid brick wall at 1000 miles per hour. |
| Visual Reasoning Cause-and-effect Relations | The model fails to correctly depict cause-and-effect relationships. | In vibrant pulp art style à la Robert McGinnis: A glamorous scientist in a 1950s-style lab coat recoils as colorful chemicals spill and mix on a cluttered lab bench. Show the immediate consequences. |
| FG-BG relations | The model has difficulty distinguishing or correctly relating foreground and background elements. | A poster about a hairpin peeking out from a discarded popcorn box. The background has a vibrant, chaotic carnival scene at night. Dazzling neon lights illuminate a bustling midway filled with towering rides, colorful game booths, and crowds of excited people. |
| Text-based | The model inaccurately generates or positions text elements in the image. | Design a logo centered around the letter S for a social network platform that connects fortunetellers with pet lovers. The S should be stylized to evoke mystical and fortune-telling themes. The overall shape should maintain the recognizability of the letter S while feeling magical and interconnected while employing animal motifs. |
| Negation | The model generates elements that negate specified details usually present in the scene. | A bustling city park with people enjoying a sunny day, but there are no trees, grass, children, or animals. Instead, the ground is covered in colorful geometric shapes and the sky is filled with floating musical instruments. Hyperrealistic and dynamic lighting. |
| Perspective | The model inaccurately represents perspective in the scene. | Cinematic close-up of an inverted birthday party hat on a wooden table, vibrant colors, soft studio lighting, 4K resolution. |
| Scaling | The model produces objects with incorrect scale. | An enormous ant, carrying a miniature skyscraper on its back. The ant stands next to a regular-sized wooden pencil for scale. By DreamWorks. |

| | | |
|---|---|---|
| Surreal | The model produces fantastical or bizarre elements when not specified. | A comical scene of a tarantula sitting at a school desk, taking an exam. The tarantula wears thick, round glasses and has a determined expression. It's staring intently at a test paper. Surrounding the tarantula are other empty desks. Bright, cartoon-style colors, bold outlines, and exaggerated expressions. Include some humorous details like a hidden cheat sheet. |
| Social Relations | The model fails to accurately depict social interactions. | An oil painting depicting an event in ancient Rome. A long table shows clear social hierarchy. The painting should capture the subtle interplay of emotions, social status, and unspoken tensions typical of the era. |
| Short Text Specific | The model inaccurately renders short text, affecting its readability. | A neon sign in a bustling city alley at night, glowing with the words 'Welcome to the City of Dreams, Open 24/7 for all your desires.' |
| Long text specific | The model inaccurately renders long specific text, affecting its readability. | A wooden signpost in a peaceful meadow, with the following inscription: "Welcome to the Land of Tranquility. Here, every step you take leads you closer to inner peace. Take a moment to breathe, relax, and let go of all your worries. Remember, in this world, you are free to be yourself and to follow the path that brings you joy." |
| Action and motion representation | The model struggles to accurately depict dynamic actions and movement. | A sequence of three images showing a person performing a cartwheel, from left to right. The first image shows the person sideways, arms raised, about to begin. The middle image captures them mid-cartwheel, legs spread wide in the air, hands planted on the ground. The final image shows them landing, other side up. |
| Anatomical limb and torso accuracy | The model generates human or animal figures with anatomically incorrect limbs or torsos. | A drawn close-up of a human hand holding a small object. The hand should be in a three-quarter view, with fingers slightly spread. Show detailed skin textures, including knuckle creases, fingernails, and subtle veins on the back of the hand. |
| Emotional conveyance | The model fails to accurately depict emotions through facial expressions or gestures. | A person standing at a podium, accepting an award with tears of joy streaming down their face, while simultaneously receiving news via an earpiece that a loved one has fallen seriously ill. Their expression should convey both elation and heartbreak. Natural light photo, photo realism, 4k, ultra realistic. |
| Tense and aspect variation | The model struggles to represent different tense or aspect variations. | A Himalayan village where climbers are preparing their gear, while a guide who has been leading expeditions for decades shares stories. In the distance, a temple that was built centuries ago glows in the morning light. Watercolor painting. |

| Tense+Text Rendering + Style | The model fails to maintain consistent tense, text placement, and style. | A vibrant urban alleyway where graffiti artists are currently painting a massive mural. A section of the wall, which was tagged with colorful graffiti last night, boldly displays the phrase 'Art is Freedom'. In the background, older layers of faded graffiti tell the story of the city's artistic evolution. Watercolor painting in the style of Paul Klee. |
|---|---|---|
| Depicting abstract concepts | The model struggles to visually represent complex, abstract concepts. | Depict the philosophical depth of religion and science, illustrating their complex relationship and profound questions about existence, truth, and the universe. Incorporate symbolism, alphabets, and numbers. Yellow monochromatic, high-resolution, aesthetic. |
| Human Action | The model generates scenarios where humans perform actions. | A person is performing a perfect handstand on a beach at sunrise, with the waves gently crashing in the background. |
| Human Anatomy Moving | The model generates scenarios where humans have natural limb movements. | A person is painting a canvas, their hands holding a palette and a brush. The background shows a creative studio filled with various art supplies and paintings. |
| Background and Foreground Mismatch | The model creates scenes where the background and foreground do not match logically. | A person working on a laptop in a jungle setting, surrounded by dense foliage, exotic animals, and a waterfall in the background. The foreground should logically blend with the natural surroundings. |
| Blending Different Styles | The model is unable to blend multiple artistic styles in one image. | A road drawn in crayon goes through a colorful photorealistic forest, with a hand-drawn pencil mountain in the background and an oil-painted sky overhead. |
| Opposite of Normal Relation | The model has a text input that is possible but unlikely or opposite of expected. | A unicorn riding a man on the moon, vibrant colors, 4k resolution. |

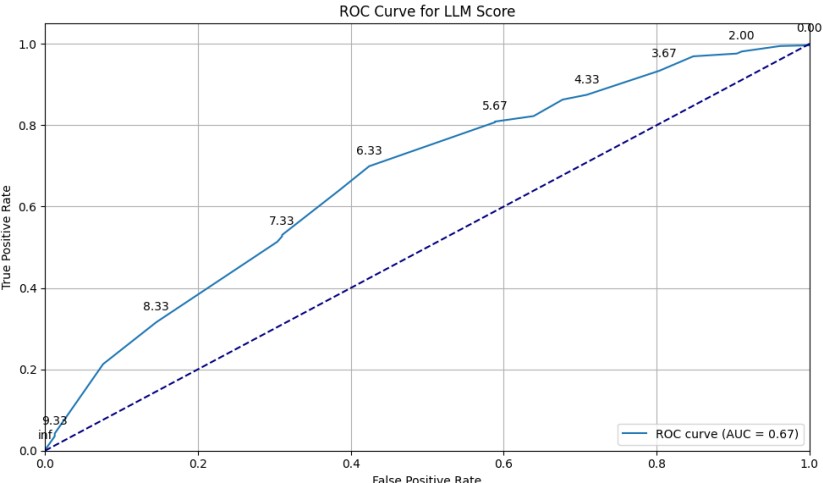

Figure 3: ROC curve depicting the performance of the LLM Judge. The curve illustrates the trade-off between true positive rate (TPR) and false positive rate (FPR) at various threshold settings.

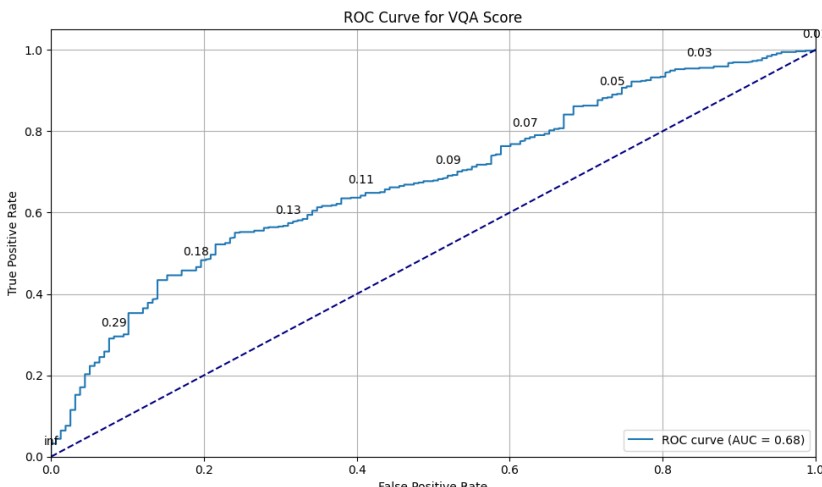

Figure 4: ROC curve for VQA Score evaluation, showing the classification effectiveness by plotting the true positive rate (TPR) against the false positive rate (FPR) for varying thresholds.

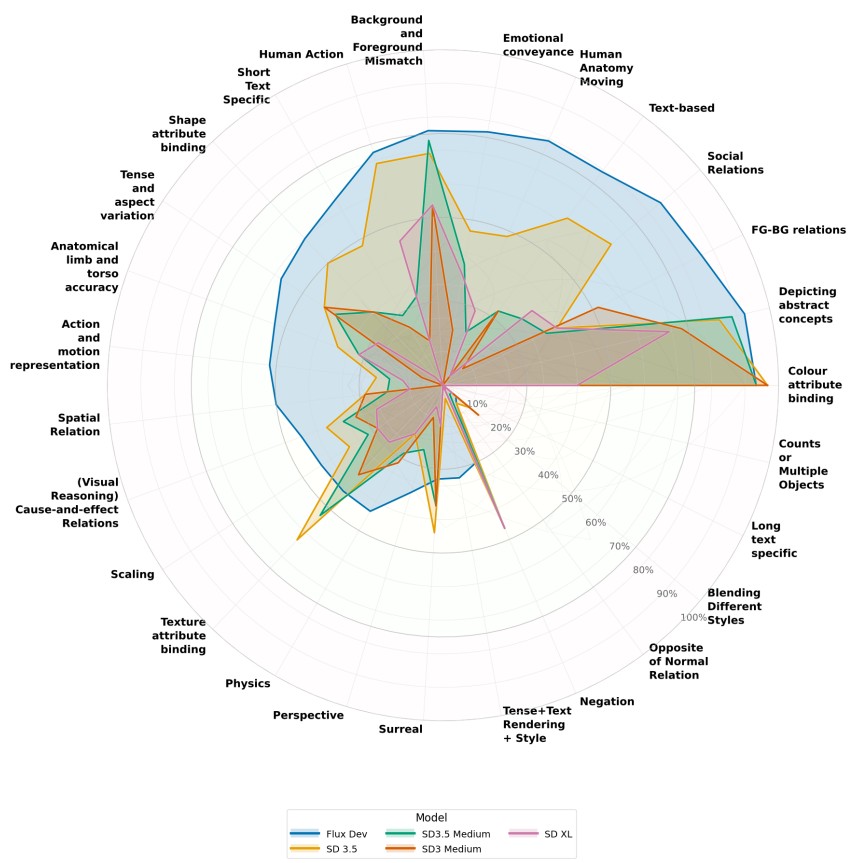

Figure 5: Star plot comparing 5 diffusion modes on 27 failure modes as graded by a human evaluation

Table 10: Tailored VLM instructions for all 27 failure modes. Each instruction is designed to query specific aspects of the failure mode to determine if the generated image matches the prompt requirements.

| Failure Mode | Sample T2I Prompt | Tailored VLM Instruction |
|---|---|---|
| Counts or Multiple Objects | An arrangement of exactly two red apples and precisely three yellow bananas on a circular plate. | Count how many apples are there? Count how many bananas are there? |
| Color Attribute Binding | A miniature red sheep driving a white car, Pixar-style 3D rendering. | What color is the sheep? What color is the car? |
| Shape Attribute Binding | A surreal landscape featuring a large, pyramid-shaped cloud floating in the sky. Below it, a circular lake reflects the cloud. | What shape is the cloud? What shape is the lake? |
| Texture Attribute Binding | An award-winning photo of a cute marble boat with visible veining, floating on a rough sea made entirely of sandpaper. | What is the texture of the boat? What is the texture of the sea? What is the texture of the sandpaper? |
| Spatial Relations | A puppy balanced precariously on the head of a patient dog, studio lighting. | Describe the spatial relations of the objects from each other. Only output the objects' spatial relations relative to one another. |
| Physics | A ball with a very low elastic modulus hitting a solid brick wall at 1000 miles per hour. | Based on the prompt, does this image show what would be the result based on natural physical laws? |
| Cause-and-effect Relations | A glamorous scientist recoils as colorful chemicals spill and mix on a lab bench. Show the immediate consequences. | Based on the prompt, does this image show what would be the result based on known cause and effect relationships? |
| FG-BG Relations | A hairpin peeking out from a discarded popcorn box with a vibrant carnival scene in the background. | Describe if any objects are blurry or out of focus. Describe if they are in the background or foreground. |
| Text-Based | Design a logo centered around the letter S for a social network platform. | What letter or letters are in the logo? What does the text say? |
| Negation | A bustling city park with people enjoying a sunny day, but there are no trees, grass, children, or animals. | Are there trees? Is there grass? Are there children? Are there animals? |
| Perspective | Cinematic close-up of an inverted birthday party hat on a wooden table. | Describe the perspective from which this scene is viewed. Is the hat inverted (upside-down)? |
| Scaling | An enormous ant carrying a miniature skyscraper on its back, next to a regular-sized pencil for scale. | Describe the relative sizes of the ant, skyscraper, and pencil. |
| Surreal | A tarantula sitting at a school desk, taking an exam, wearing glasses. | Describe any surreal or fantastical elements in the image. |
| Social Relations | An oil painting depicting a feast in ancient Rome showing clear social hierarchy at a long table. | Describe the social dynamics and hierarchy visible in the image. |
| Short Text Specific | A neon sign in a city alley: 'Welcome to the City of Dreams, Open 24/7 for all your desires.' | What does the text say? Is it readable and accurate? |
| Long Text Specific | A wooden signpost with: "Welcome to the Land of Tranquility. Here, every step..." | What does the text say? Is all the text readable and accurate? |
| Action and Motion | A sequence of three images showing a person performing a cartwheel from left to right. | Describe the action being performed. Is the motion sequence logical? |
| Anatomical Accuracy | A close-up of a human hand holding a small object with detailed skin textures. | Are there any anatomical deformities? Is the hand anatomically correct with proper finger count? |
| Emotional Conveyance | A person at a podium with tears of joy but receiving tragic news via earpiece. | Describe the emotions conveyed in the facial expression. |
| Tense and Aspect | A Himalayan village where climbers are preparing gear, while a guide shares stories. | Describe the temporal aspects. What activities are ongoing vs. completed? |
| Text Rendering + Style | An urban alley where artists are painting a mural with 'Art is Freedom' in the style of Paul Klee. | What does the text say? What artistic style is used? |
| Abstract Concepts | Depict the philosophical depth of religion and science with symbolism, alphabets, and numbers. | Is this prompt being represented? Describe the image and its themes. |
| Human Action | A person performing a perfect handstand on a beach at sunrise. | What actions is the person performing? Are there any anatomical deformities? |
| Human Anatomy Moving | A person painting a canvas, hands holding a palette and brush. | What actions is the person doing? Are the hands anatomically correct with proper finger count? |
| BG-FG Mismatch | A person working on a laptop in a jungle setting surrounded by exotic animals. | Is there a person in the foreground and jungle in the background? Are they anatomically correct? |
| Blending Styles | A crayon road through a photorealistic forest with pencil mountains and oil painted sky. | What is crayon? What is photorealistic? What is pencil? What is oil painted? |
| Opposite Relation | A unicorn riding a man on the moon, vibrant colors. | What is the relation between the man and the unicorn in the image? |

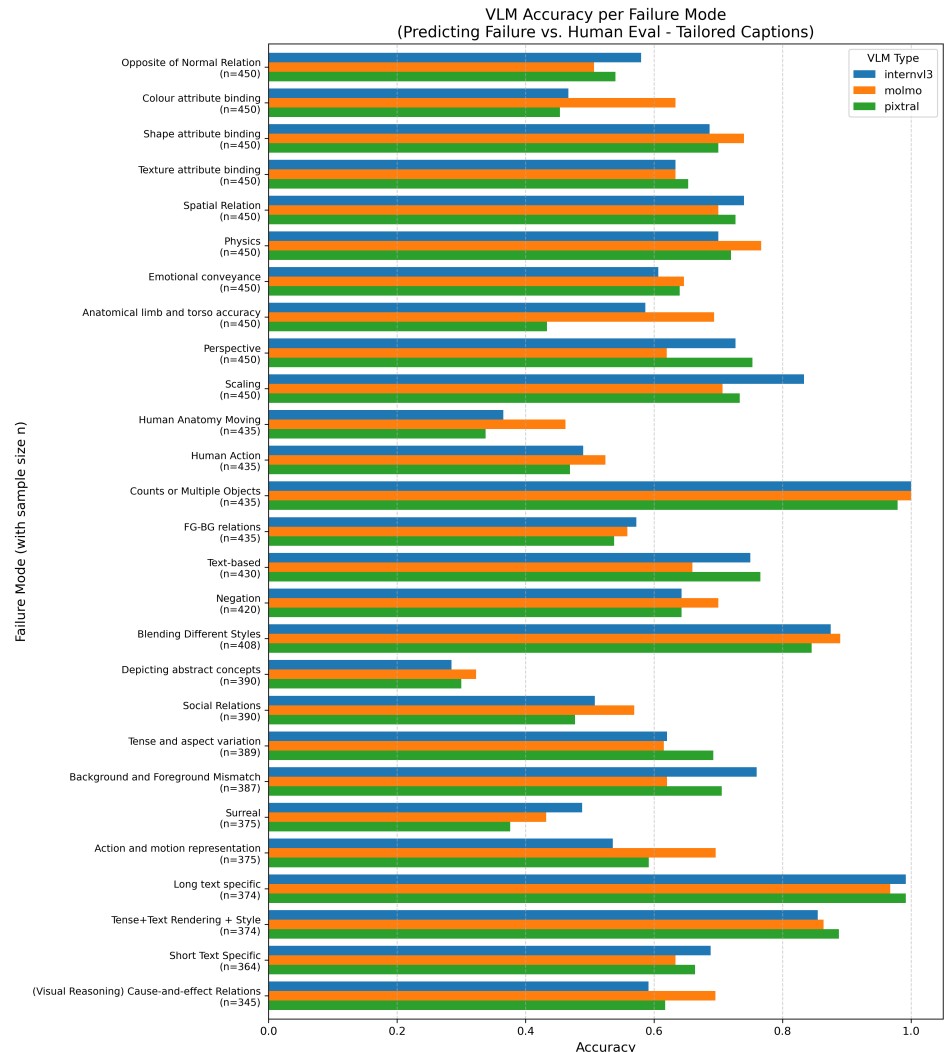

Figure 6: Comparison of VLM Accuracy Across Different Failure Modes. Accuracy is defined as the VLM's predicted failure boolean matching the human evaluation boolean for tailored captions. Average accuracy across all failure modes: Molmo-72B 76.5±1.5%, InternVL3-78B 75.5±1.6%, Pixtral-124B 74.2±1.6% (n=750 prompts per model). The three VLMs demonstrate comparable performance with no statistically significant differences in most pairwise comparisons.

