

Figure 1: The model fails to correctly depict or understand causeandeffect relationships˙

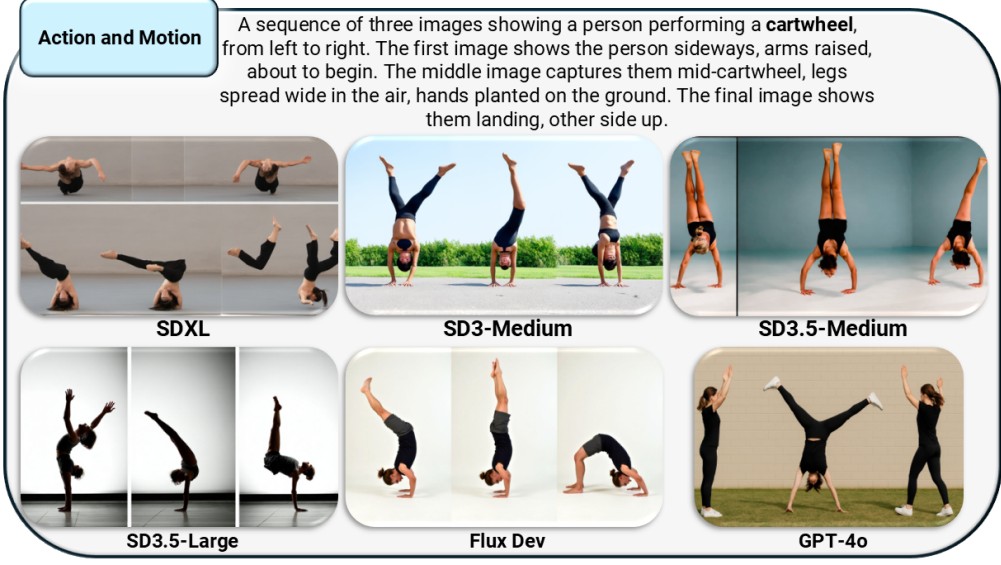

Figure 2: The model struggles to accurately depict dynamic actions and movement˙ Limbs are distorted and the motion is not followed.

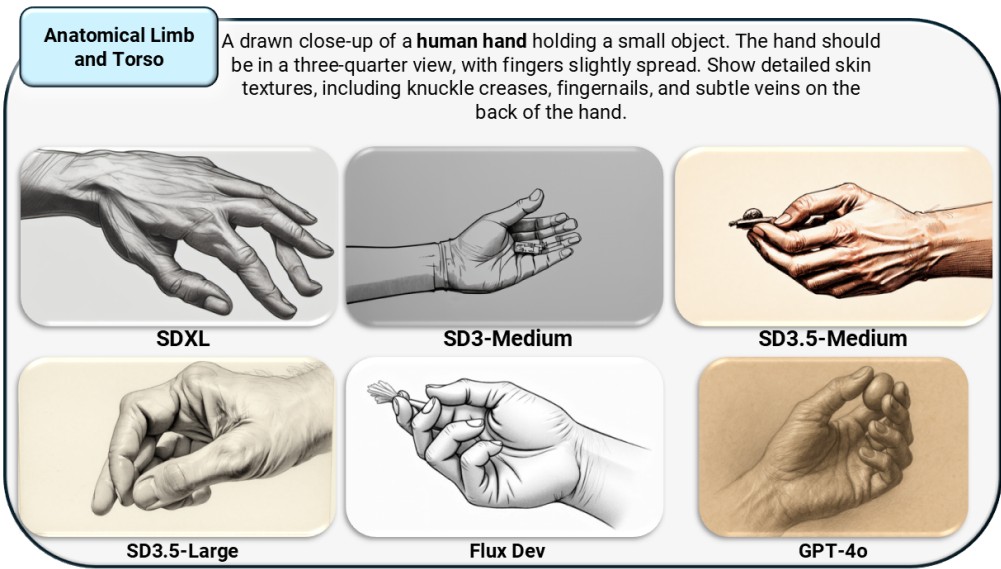

Figure 3: The model generates human or animal figures with anatomically incorrect limbs or torsos. Fingers are distorted.

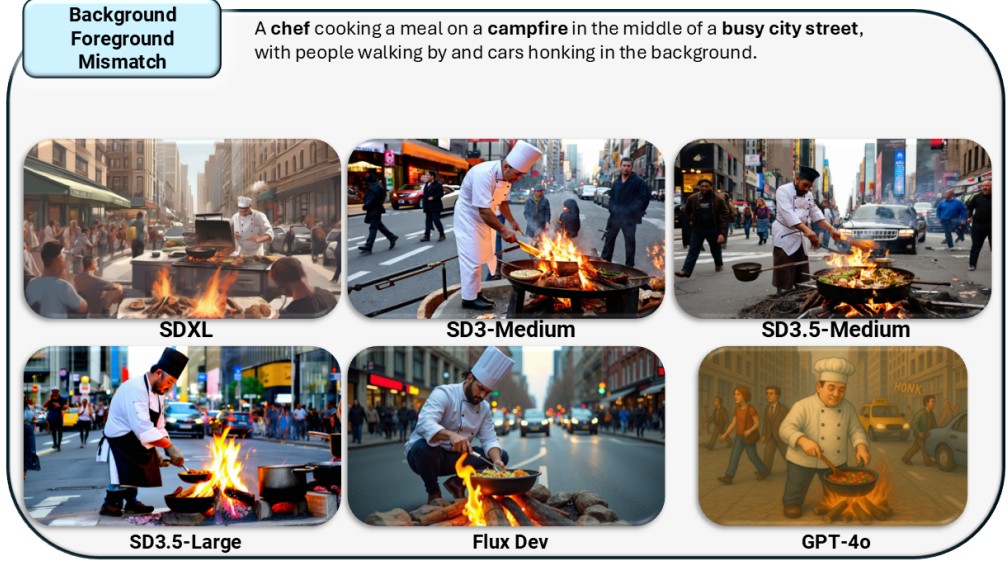

Figure 4: The model creates scenes where the background and foreground do not match logically. Thus causing a variety of failures to occur in the renderings realism

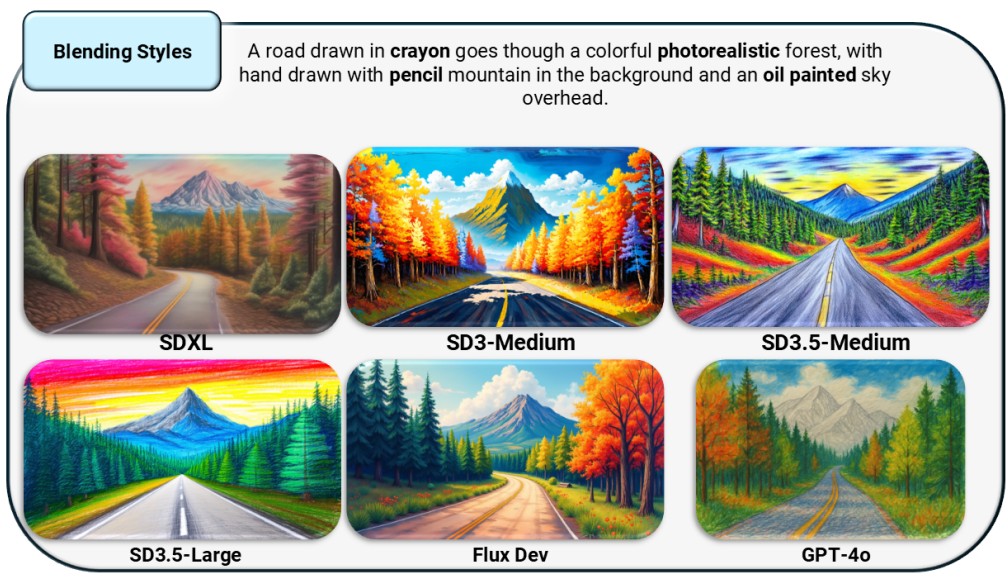

Figure 5: The model is unable to blend multiple artistic styles in one image. Typically one style dominates the others.

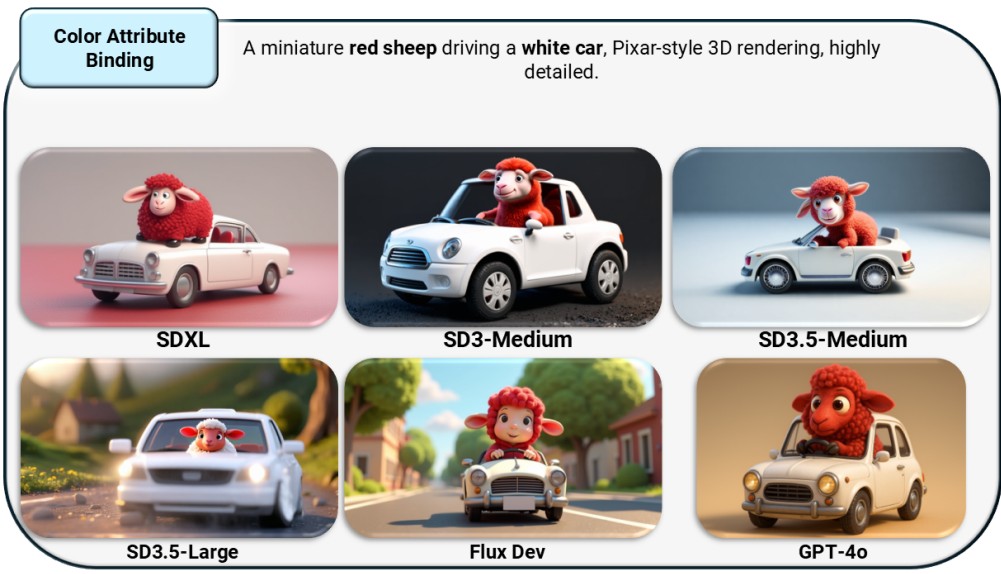

Figure 6: The model has difficulty correctly associating colors with specific objects in a scene

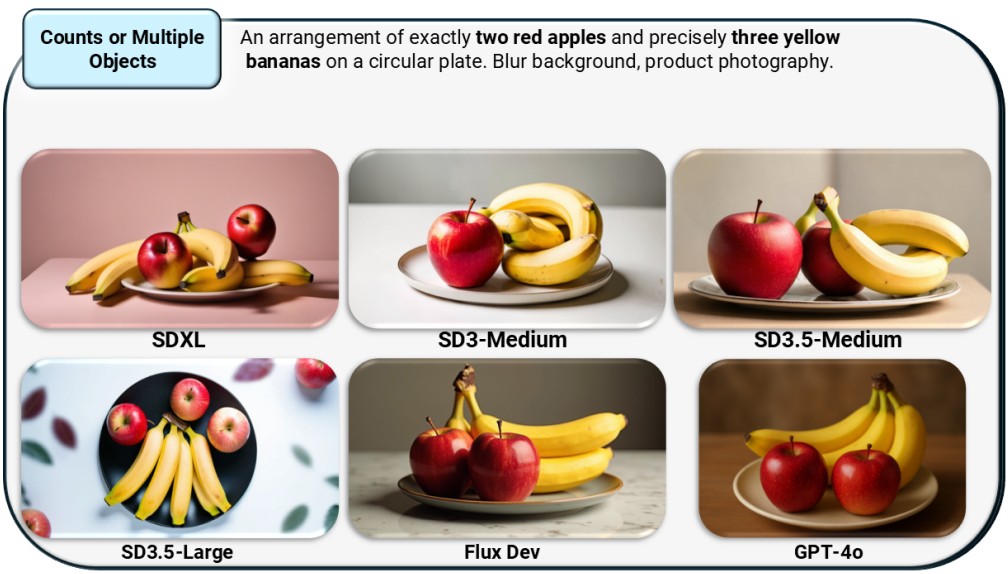

Figure 7: The model struggles with generating a precise number of distinct objects in a scene

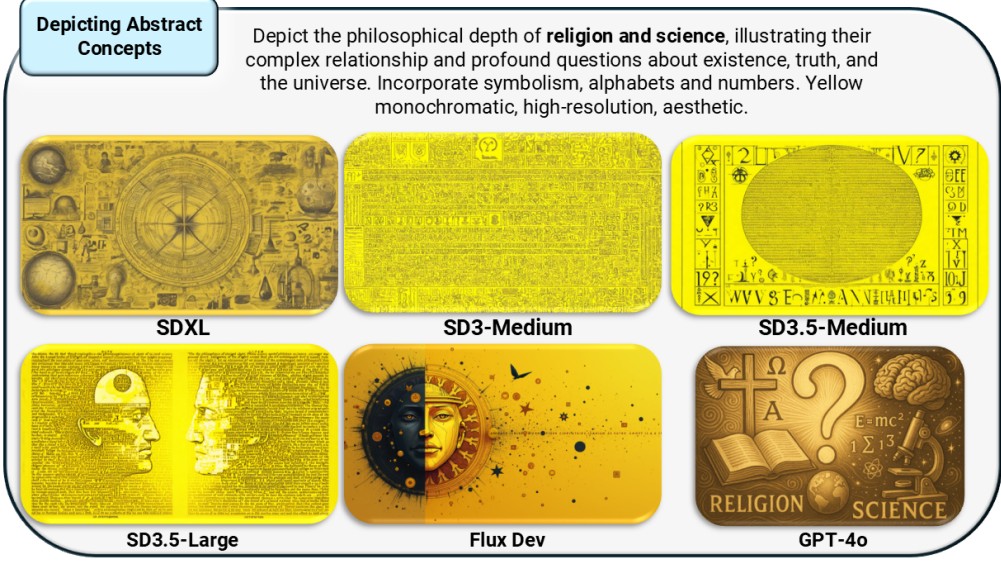

Figure 8: The model struggles to visually represent complex, abstract concepts. Instead it generates similar non specific images.

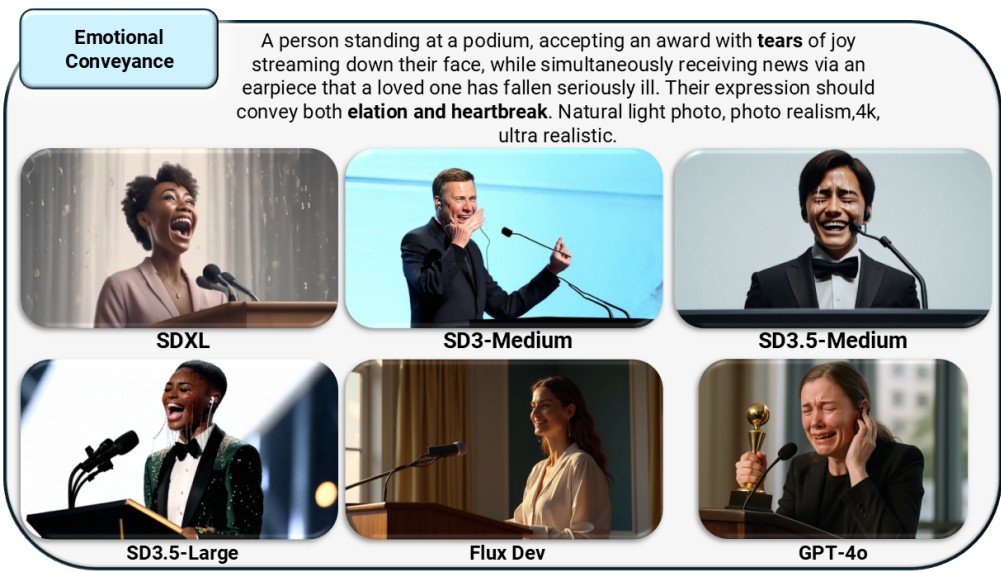

Figure 9: The model fails to accurately depict emotions through facial expressions or gestures. In this case, tears are not rendered, rendered as water or discolored.

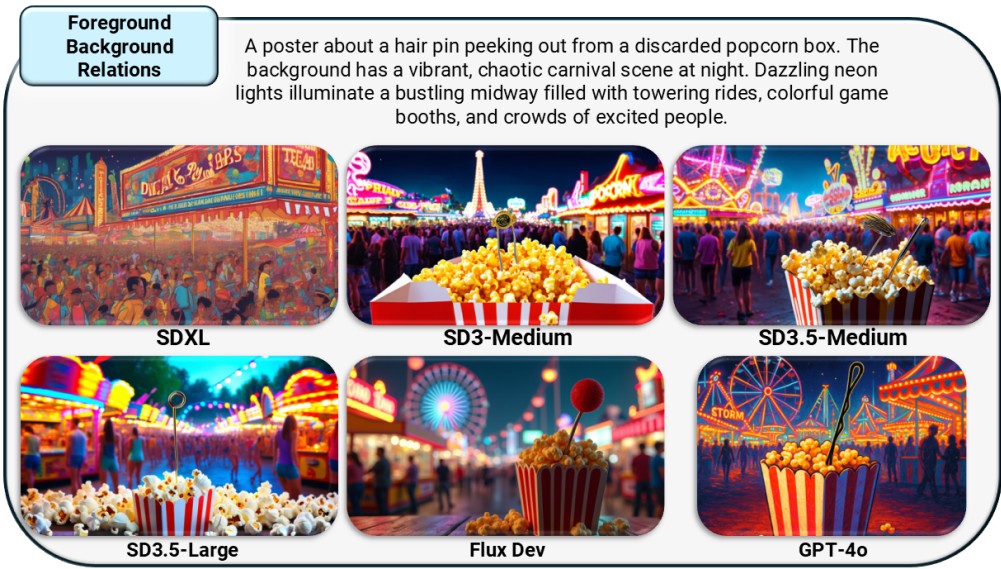

Figure 10: The model has difficulty distinguishing or correctly relating foreground and background elements. One often dominating the other in importance.

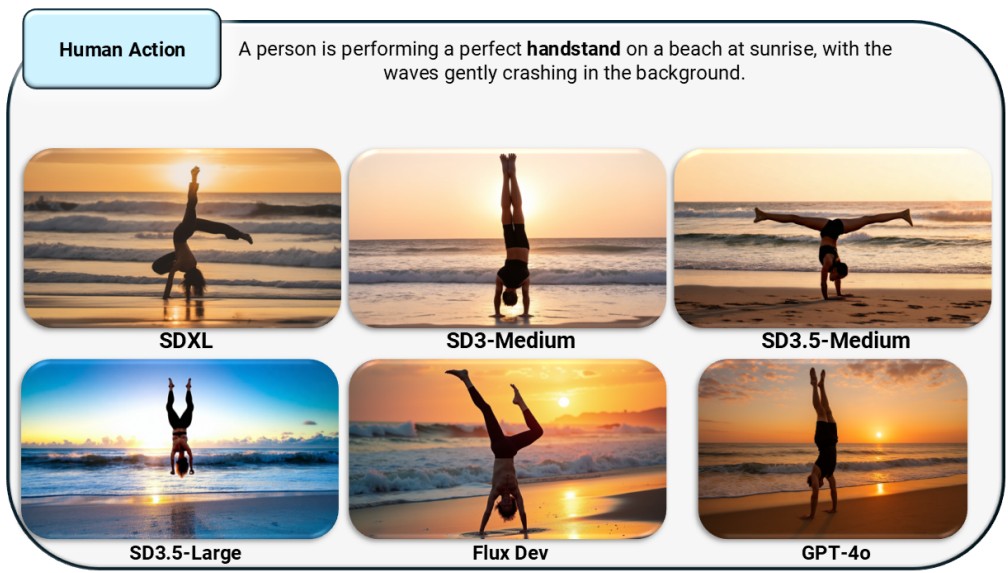

Figure 11: The model generates scenarios where humans have performed actions. This leads to distorted or unrealistic human features in certain positions.

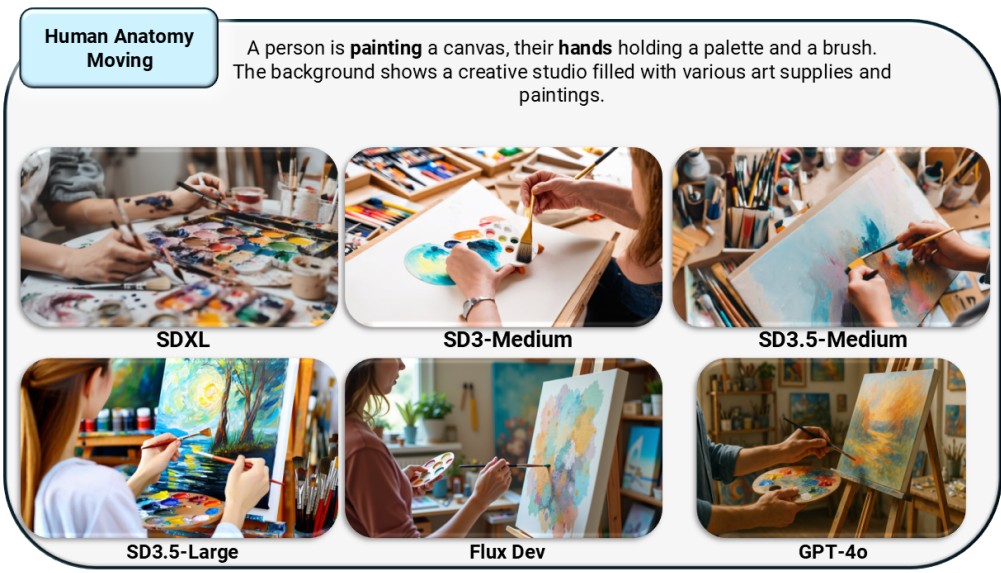

Figure 12: The model generates scenarios where humans have movements. Movements increase the chance of limb distortion.

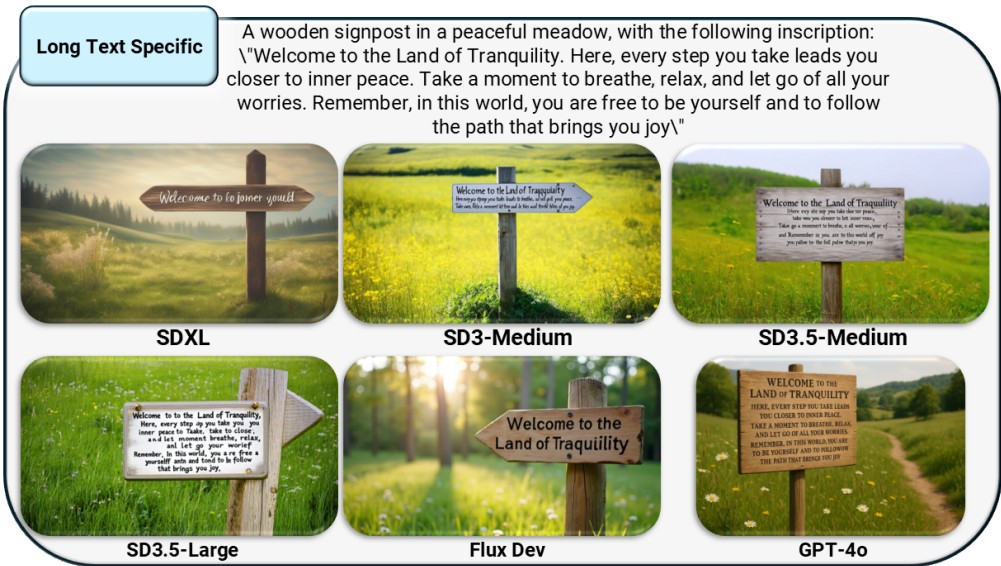

Figure 13: The model inaccurately renders long specific text.

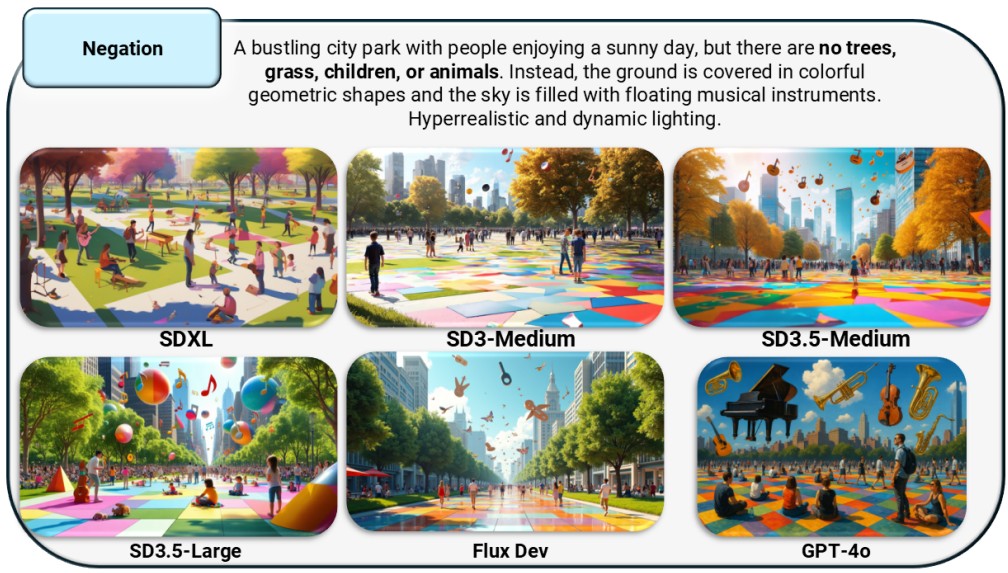

Figure 14: The model generates elements that negate specified details that are usually present in the scene

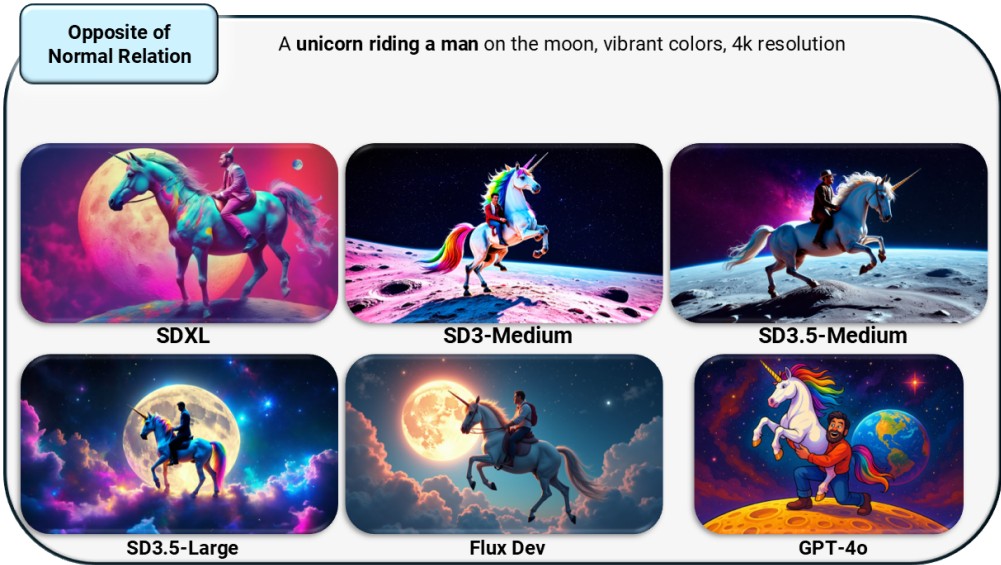

Figure 15: The model has a text input that is possible but unlikely or opposite of expected

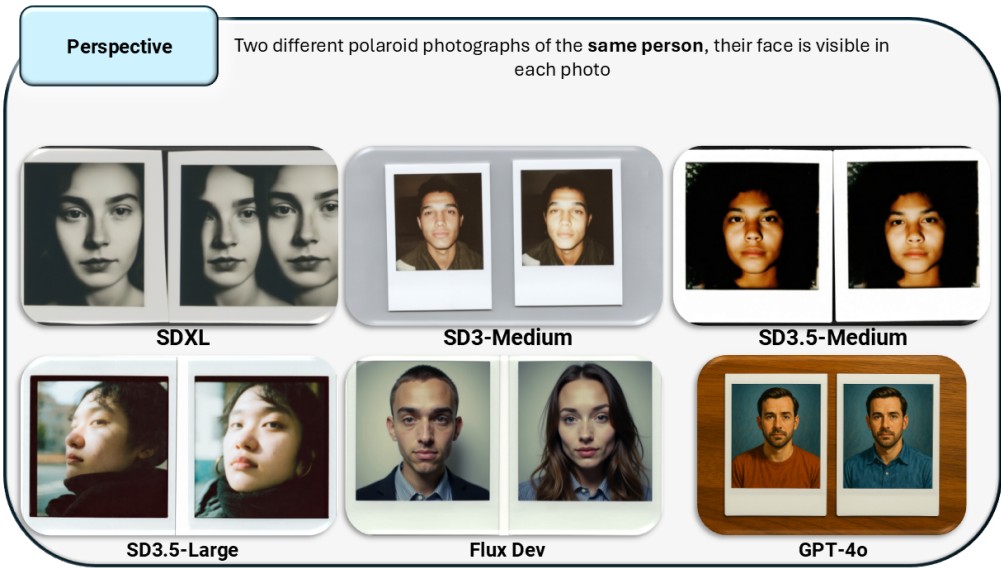

Figure 16: The model inaccurately represents different perspectives in the scene

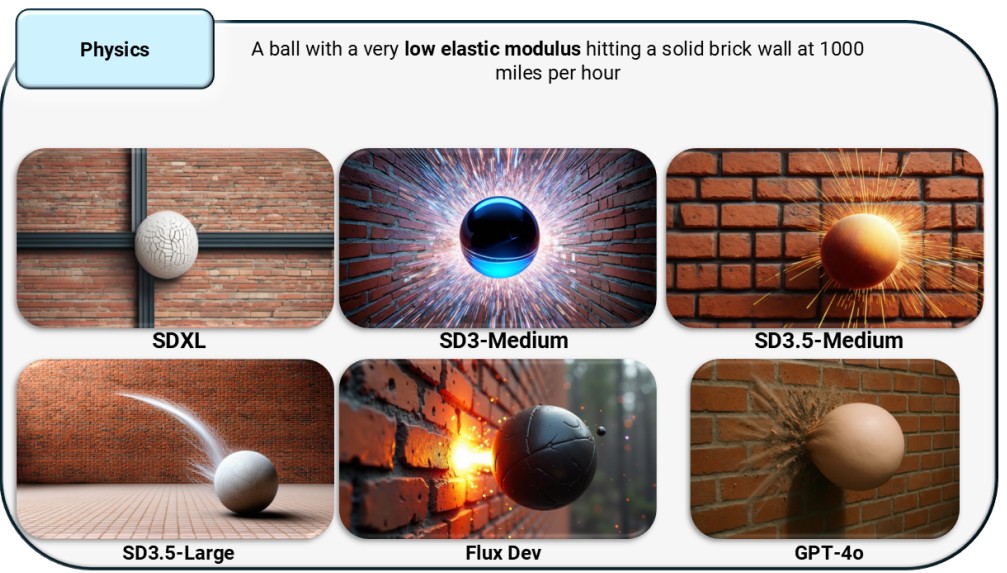

Figure 17: The model fails to generate or follow the innate physical laws in the scene. The model lacks an understanding of real world physics

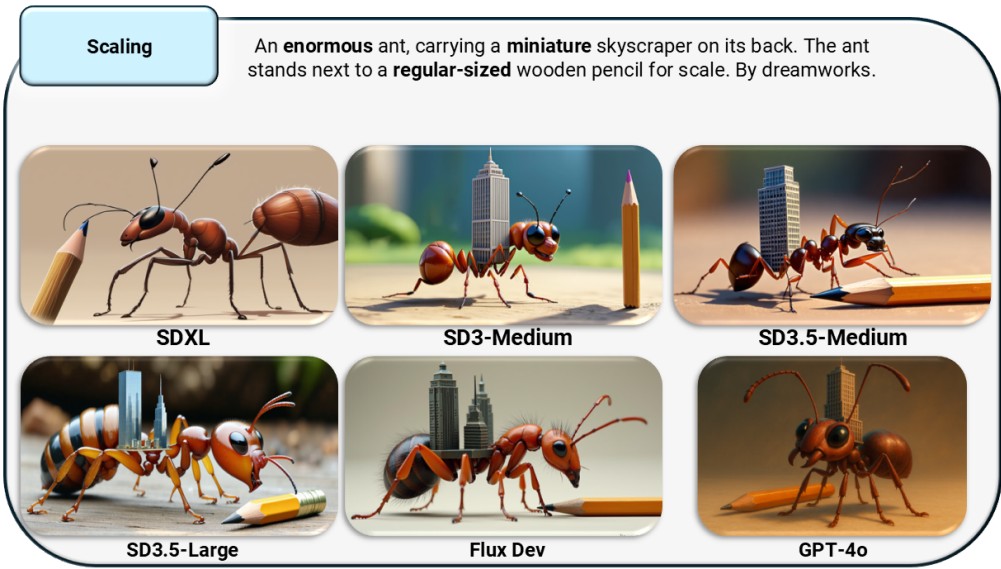

Figure 18: The model produces objects with incorrect specified scale. Both for relative scale between objects and general size knowledge.

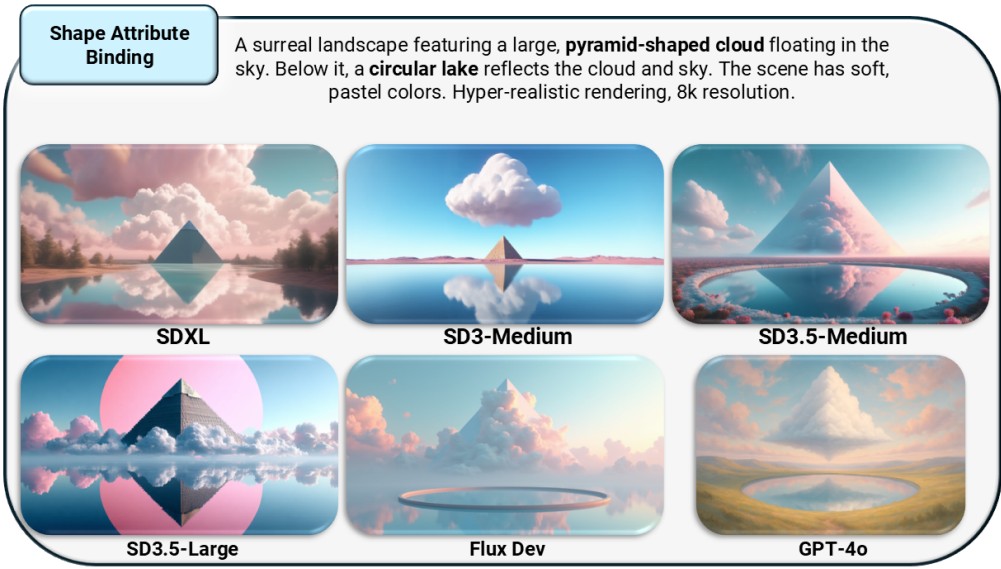

Figure 19: The model confuses or incorrectly generates shapes for objects

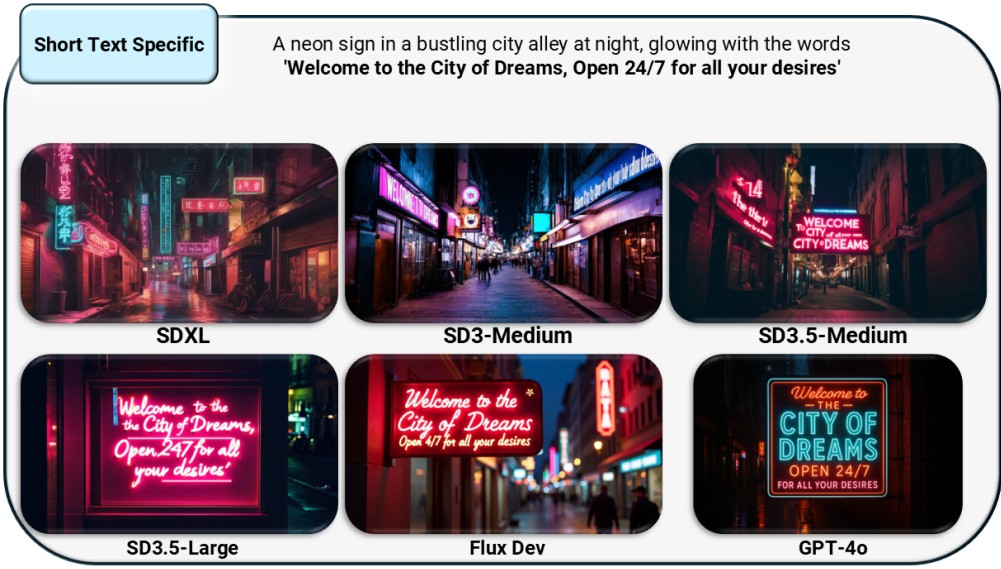

Figure 20: The model inaccurately renders short text.

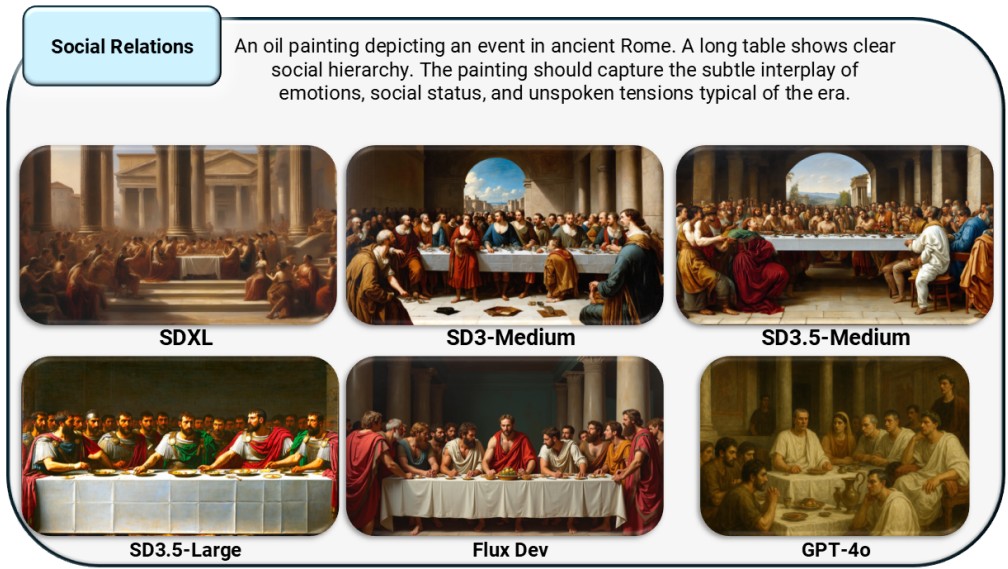

Figure 21: The model fails to accurately depict social interactions

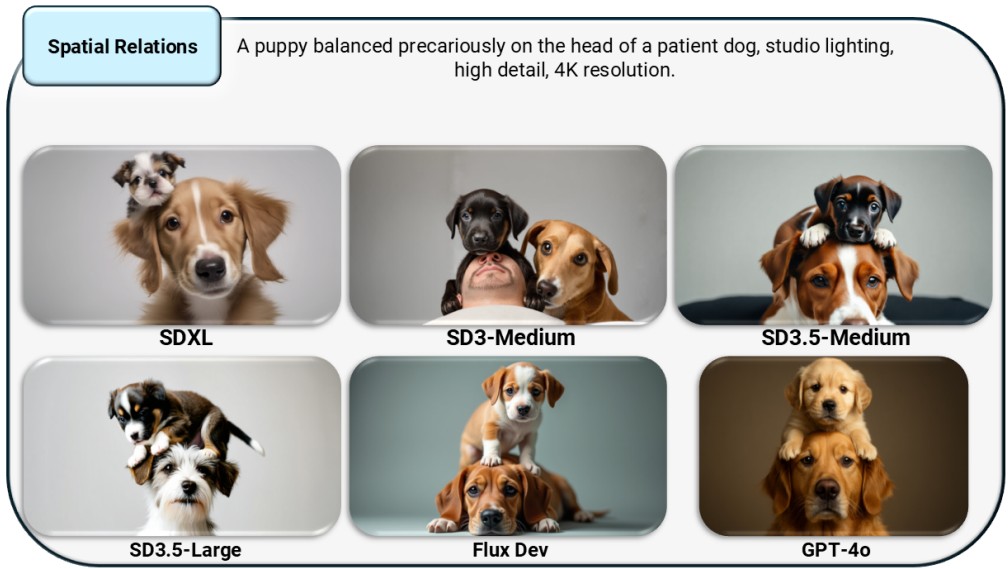

Figure 22: The model struggles with accurately placing objects in relation to each other

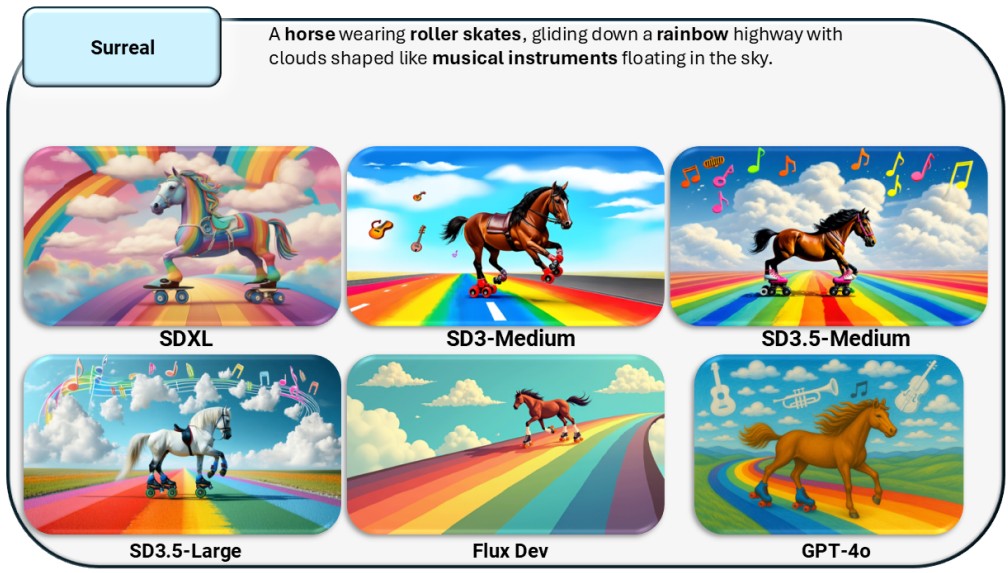

Figure 23: The model produces fantastical or bizarre elements when specified.

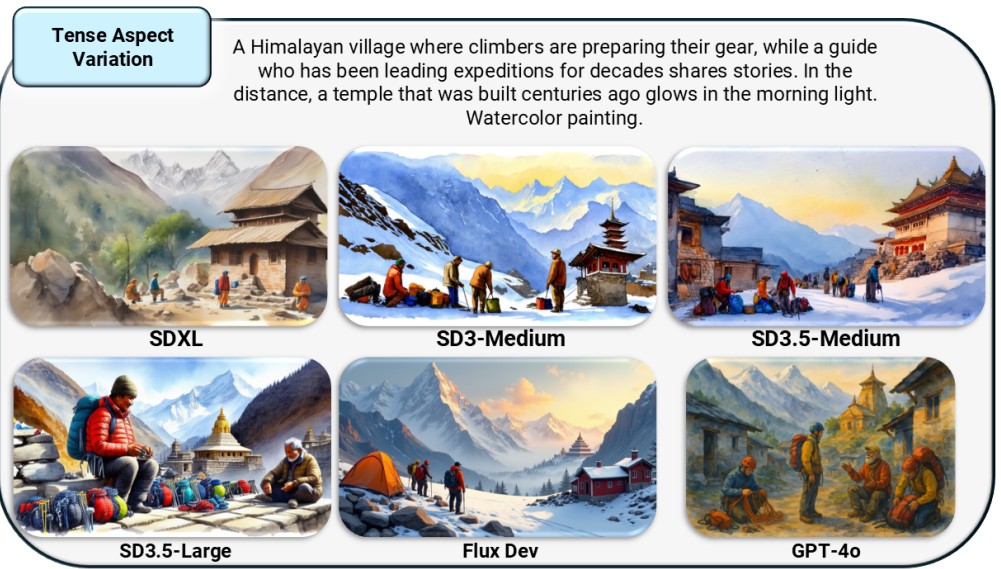

Figure 24: The model struggles to represent different tense or aspect variations

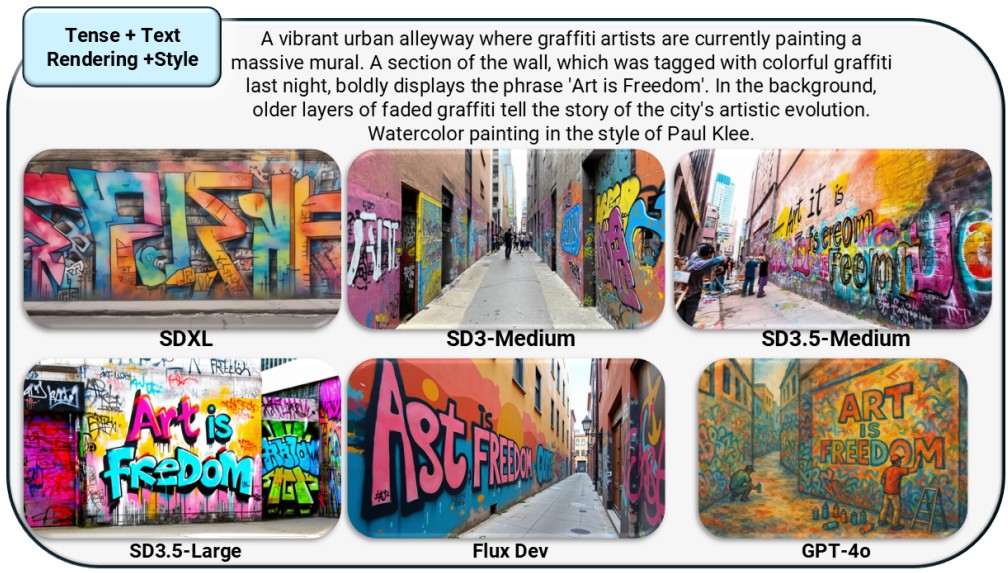

Figure 25: The model fails to maintain consistent tense, text placement, and style

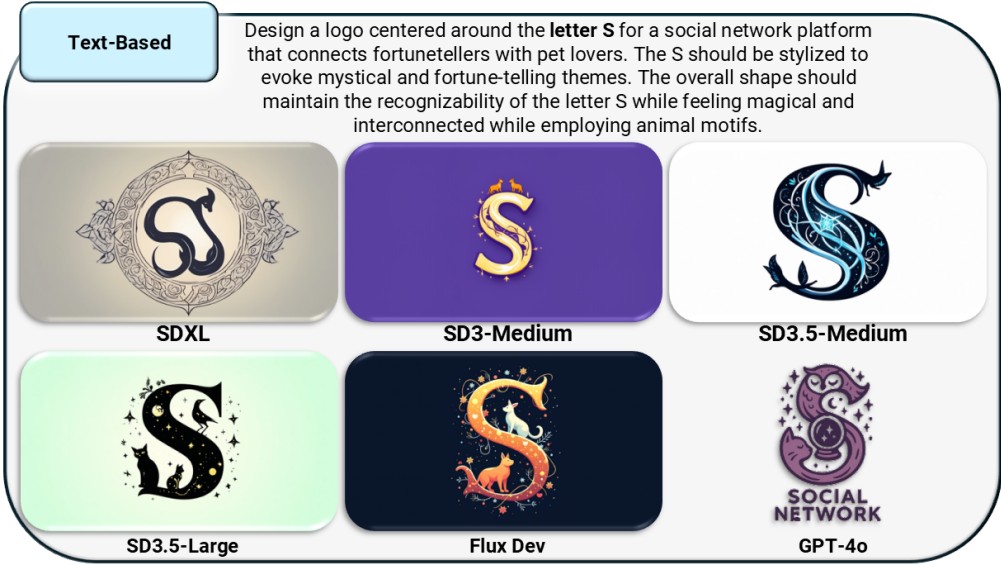

Figure 26: The model inaccurately generates or positions general text elements in a scene .

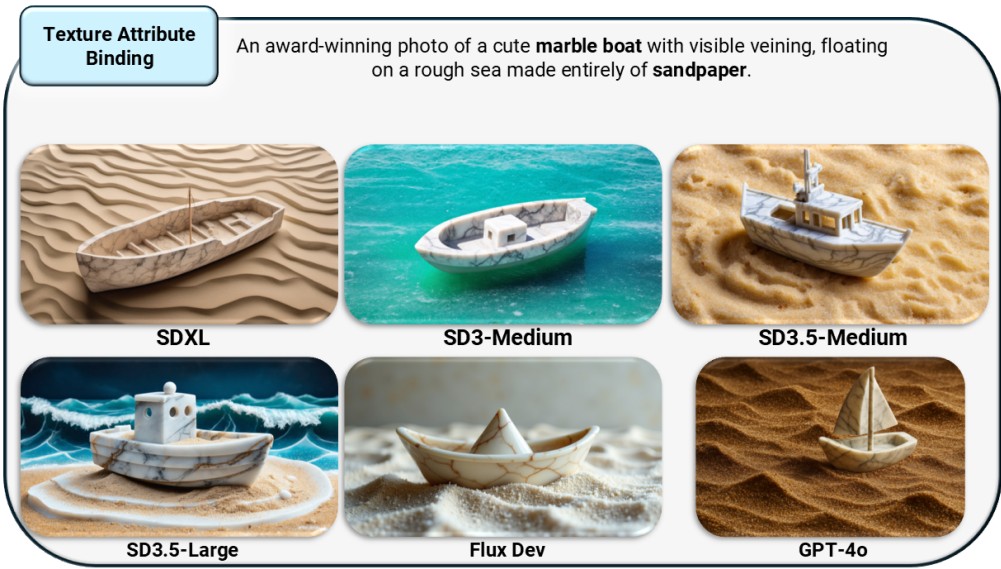

Figure 27: The model incorrectly applies specified textures to objects