# OpenReview forum: "FineGRAIN: Evaluating Failure Modes of Text-to-Image Models with Vision Language Model Judges"
_NeurIPS.cc/2025/Datasets_and_Benchmarks_Track — NeurIPS 2025 Datasets and Benchmarks Track spotlight_

### Official Review · Reviewer_si8o · 2025-06-23

**Rating:** 4
**Confidence:** 3

**Summary:**

The paper introduces a structured evaluation framework named FineGRAIN, designed to identify and analyze specific failure modes of text-to-image (T2I) models using vision-language models (VLMs). This framework tests T2I models’ adherence to complex user prompts, specifically examining 27 categorized failure modes like incorrect object counting, color misalignment, and inappropriate attribute bindings. The authors provide a curated dataset generated by five diffusion-based T2I models (Flux, SD3-Medium, SD3-Large, SD3.5-Medium, SD3.5-Large), annotated by a VLM (Molmo), with scoring by a language model (Llama3). FineGRAIN produces boolean evaluations, detailed explanations, and rankings that highlight systematic errors and reveal nuanced capabilities and limitations of current generative models.

**Dataset Code Accessibility:**

Yes

**Ethical Considerations:**

No, there are no or only very minor ethics concerns

**Final Justification:**

Thank you for the author's rebuttal, which has partially addressed my concerns. Starting from failure modes is an interesting and worthwhile direction to explore. Meanwhile, I also hope to see more modes being located. Additionally, I would like to see how they can feed back into the generative models. It is recommended that the authors supplement the content of the rebuttal into the main text. I believe this work is of certain significance and promoting effect to the community, so I will raise my score from 3 to 4.

**Limitations Weaknesses:**

1.	There is insufficient discussion and comparative experimentation regarding the advantages of this work over previous evaluation metrics, making it difficult to quantify its value and significance.
2.	The purported advantages of jointly evaluating Vision-Language Models (VLMs) and Generative Models appear to be unclear. When the performance of VLMs is suboptimal, it remains uncertain how the proposed FineGRAIN approach can be effectively utilized.
3.	It is also unclear how FineGRAIN can be employed to enhance existing text-to-image (T2I) models.
4.	The evaluation pipeline relies heavily on the accuracy of VLM (Molmo) and LLM (Llama3). Any inherent biases or inaccuracies in these models could affect the integrity of the FineGRAIN framework’s outcomes.
5.	The use of large-scale VLMs and LLMs, while effective, introduces significant computational costs. The paper acknowledges but does not deeply explore how this impacts the feasibility of wide adoption or large-scale testing.
6.	The paper does not sufficiently explore statistical significance, confidence intervals, or provide error bars for experimental results, making it harder to interpret reliability and generalizability of findings.

**Strengths Contributions:**

1.	FineGRAIN provides a comprehensive and nuanced method of identifying model errors beyond traditional aggregate metrics like CLIPScore and VQAScore, offering detailed insights into specific strengths and weaknesses.
2.	The introduction of 27 distinct, precisely defined failure modes allows targeted analysis and detailed diagnosis of errors, surpassing previous benchmarks that generally only addressed broader failure modes.
3.	Evaluating both T2I and VLM simultaneously enriches the analysis and allows for more robust conclusions about model capabilities and failure interactions.

---

> ### Author Rebuttal · Authors · 2025-07-31
>
> Thank you for your detailed review of our work. We appreciate your recognition of our benchmark surpassing previous benchmarks and that evaluating T2I and VLM simultaneously enriches the analysis and allows for more robust conclusions about model capabilities and failure interactions in your review.
>
> 1. >There is insufficient discussion and comparative experimentation regarding the advantages of this work over previous evaluation metrics, making it difficult to quantify its value and significance.
>
> **Our evaluation method and benchmark is more extensive than previous state-of-the-art evaluation methods as shown in table 3.** We focus our comparisons to CLIPScore and VQAScore as we can directly apply those metrics to our dataset. Ideally, we would do a one-to-one comparison of all methods both on our dataset and theirs however given **many benchmark evaluations have been tailored to their datasets** means of evaluation it is not always a fair comparison. Even with VQAScore we had to rely on some ROC analysis. That said, we did show our benchmark scores were better than VQAScore on failure modes like text rendering, which the T2I community is particularly interested in.
>
> 2. >The purported advantages of jointly evaluating Vision-Language Models (VLMs) and Generative Models appear to be unclear. When the performance of VLMs is suboptimal, it remains uncertain how the proposed FineGRAIN approach can be effectively utilized.
>
> By jointly evaluating the failure modes of VLM’s and T2I models we can better understand the failure modes in Multimodel vision as a whole, thus why we conducted the manual human evaluation of all +3750 images. We agree that the pipeline will underperform when the VLM underperforms, **hence why it was so important that we also evaluate its reliability**, thus being our own critic to our benchmark evaluation method. Lastly, this joint evaluation helps the community understand where we are in terms of potential VLM reward modeling for future improvement to T2I models by the failure modes of T2I models.
>
> 3. >It is also unclear how FineGRAIN can be employed to enhance existing text-to-image (T2I) models.
>
> Part of our goals in this project was to evaluate the feasibility of using VLM’s as a best-of-N reward model for T2I models. It is a **natural progression from the joint evaluation for well scoring VLM failure modes.** At our present evaluation, failure modes in which both VLM’s and T2I models overlap present a challenge for either model to act as a critic of the other, however failure modes in which they differ have a potential to allow one model to teach the other. Due to the challenges in implementing such a pipeline we extensively evaluate T2I and VLM failure modes to understand where this may be applicable and possibly how to mitigate challenges like reward hacking.
>
> 4. >The evaluation pipeline relies heavily on the accuracy of VLM (Molmo) and LLM (Llama3). Any inherent biases or inaccuracies in these models could affect the integrity of the FineGRAIN framework’s outcomes.
>
> Yes, this is an inherent tradeoff with an automatic evaluation such as ours, as well as other contemporary T2I evaluations. This is also in part **what prompted us to jointly evaluate the models,** including 3 VLM models (Molmo, Pixtral, InternVL3) to understand the limitations of both our technique and current state of Multimodel vision as a whole.
>
> 5. >The use of large-scale VLMs and LLMs, while effective, introduces significant computational costs. The paper acknowledges but does not deeply explore how this impacts the feasibility of wide adoption or large-scale testing.
>
> We evaluated some of the largest open source models for our final run, as they were the highest performing. We do understand that many researchers may not have access to the computational resources to run these models locally, and thus perform the extensive unit testing, exploration and many iterations it takes to converge to this benchmark dataset. However, to only use the final form we are releasing to reproduce the results is much more **accessible as there are API’s for large VLM’s that cost less than $10 USD** to evaluate and reproduce our results given the size of our dataset. Well within the means of most researchers.
>
> 6. >The paper does not sufficiently explore statistical significance, confidence intervals, or provide error bars for experimental results, making it harder to interpret reliability and generalizability of findings.
>
> More statistical analysis or increasing the number of samples would make our approach more rigorous. The overall evaluation and benchmark is a statistically significant evaluation for state-of-the-art open source models. In pairwise comparisons, 9 out 10 comparisons are statistically significant at a $p < 0.05$ for n = 750 prompts per model. Overall significance of a chi square test $\chi^2 = \sum_{i=1}^{5} \frac{(O_i - E_i)^2}{E_i} = 134.374$, $df = 4$, $p < 0.001$ for differing from an expected mean, goodness-of-fit, which is a higher bar than a test of independence. The VLM's were much more close in their evaluation, however the evaluation still benchmarks their failure modes.
>
>
>
> | Model | Average (%) | Std Error (%) |
> |-------|-------------|---------------|
> | Flux | 51.04 | ±1.83 |
> | SD3.5 | 40.06 | ±1.79 |
> | SD3.5 M | 30.56 | ±1.68 |
> | SD3 M | 24.27 | ±1.57 |
> | SD3 XL | 21.09 | ±1.49 |
>
> | Comparison | Difference (%) | Z-score | P-value |
> |------------|----------------|---------|---------|
> | Molmo vs InternVL3 | 1.00 | 0.912 | 0.362 |
> | Molmo vs Pixtral | 2.34 | 2.124 | **0.034** |
> | InternVL3 vs Pixtral | 1.34 | 1.212 | 0.226 |

---

> > ### Comment · Reviewer_si8o · 2025-08-01
> > **Official Comment by Reviewer si8o**
> >
> > Thank you for the author's rebuttal, which has partially addressed my concerns. Starting from failure modes is an interesting and worthwhile direction to explore. Meanwhile, I also hope to see more modes being located. Additionally, I would like to see how they can feed back into the generative models. It is recommended that the authors supplement the content of the rebuttal into the main text. I believe this work is of certain significance and promoting effect to the community, so I will raise my score from 3 to 4.

---

> > ### Author Response · Authors · 2025-08-01
> >
> > Thank you for reviewing our rebuttal and raising your score. We are happy to hear that you agree that starting from failure modes is an interesting and worthwhile direction to explore and that it will have a promoting effect on the community. We also look forward to exploring more failure modes and experimenting with their use in improving generative models. We will supplement the content of the rebuttal into the main text as per your recommendation.

---

### Official Review · Reviewer_LZYw · 2025-07-02

**Rating:** 6
**Confidence:** 2

**Summary:**

To identify the limitations of text-to-image (T2I) models and vision language models (VLMs), the paper proposed a curated dataset and benchmark. It defined 27 failure modes and generate 25~30 prompts per failure mode. For validation, the paper provided images generated by five T2I models. Using these images, VLM instructions, a VLM and LLM, it validated whether a VLM identified the failure modes. The proposed benchmark exceeded the previous metrics in identifying failure modes.

**Additional Feedback:**

While I am not requesting additional experiments, it may be worth investigating whether fine-tuning the VLM could lead to further improvements in benchmark accuracy.

**Dataset Code Accessibility:**

Yes

**Ethical Considerations:**

No, there are no or only very minor ethics concerns

**Final Justification:**

Since my concern in terms of the fairness of the experimental comparisons are solved, I raised the score.

**Limitations Weaknesses:**

- It seems that the paper does not provide dataset statistics, such as the dataset size.
- The resolution of images generated from employed T2I models are missing. Were the values of resolution consistent across all models? If this was not the case, the fairness of the experimental comparisons may be compromised.

**Strengths Contributions:**

- The motivation of this work is clear.  Identifying the limitations of each T2I models is important. The contribution of this work helps not only research community but also developers to utilize T2I models.
- The paper provides experimental justification for their use of Llama3-70B and Molmo-72B. This helps readers to understand why these models are employed.
- It also provides enough experiments to evaluate the proposed benchmark.

---

> ### Author Rebuttal · Authors · 2025-07-31
>
> Thank you for reviewing our benchmark paper and recognizing in it this works contribution to the research community in said review.
>
> 1. >It seems that the paper does not provide dataset statistics, such as the dataset size.
>
> We focused on the number of failure modes and prompts per failure mode; however we can include the number of images in the camera-ready version as we did on the github and huggingface repos. Our results are statistically significant. In pairwise comparisons, 9 out 10 comparisons are statistically significant at a $p < 0.05$. Overall significance of a chi square test **$\chi^2 = 134.374$, $df = 4$, $p < 0.001$** for differing from an expected mean, goodness-of-fit, which is a higher bar than a test of independence.
>
>
> | Model | Average (%) | Std Error (%) |
> |-------|-------------|---------------|
> | Flux | 51.04 | ±1.83 |
> | SD3.5 | 40.06 | ±1.79 |
> | SD3.5 M | 30.56 | ±1.68 |
> | SD3 M | 24.27 | ±1.57 |
> | SD3 XL | 21.09 | ±1.49 |
>
> 2. >The resolution of images generated from employed T2I models are missing. Were the values of resolution consistent across all models? If this was not the case, the fairness of the experimental comparisons may be compromised.
>
> This is a good point, **all generated images are 1360 x 768** so we hadn’t thought to include it. With the exception of the GPT4 unit tests in the appendix however, these are not used for our metrics.
>
> ### Additional Feedback
>
> >While I am not requesting additional experiments, it may be worth investigating whether fine-tuning the VLM could lead to further improvements in benchmark accuracy.
>
> Thank you for the suggestion of fine-tuning the VLM. This is something that we have internally discussed, having specific VLM’s for each failure mode. Though, this may require a large amount of generated data, more computational resources and limit the accessibility of other researchers for total evaluation, given there are 27 failure modes but could be useful to focus on specific failure modes.

---

> > ### Comment · Reviewer_LZYw · 2025-08-01
> > **Responce to the rebuttal**
> >
> > Dear authors,
> >
> > Thank you for addressing my concerns.
> > Since my concerns are addressed, I raise my score.

---

> > > ### Author Response · Authors · 2025-08-01
> > >
> > > Thank you for reviewing our rebuttal and raising your score. We were happy to hear that it addressed your concerns.

---

### Official Review · Reviewer_ftKD · 2025-07-02

**Rating:** 5
**Confidence:** 3

**Summary:**

This paper proposes a novel benchmark, FineGRAIN, for evaluating both text-to-image (T2I) generative models and vision-language models (VLMs). To enable fine-grained failure evaluation, the authors define a taxonomy of 27 specific failure modes and manually construct 25–30 targeted prompts for each category. The dataset is built by generating images using open-source T2I models based on these prompts, with each image annotated by humans with a binary label indicating whether the failure is present. FineGRAIN introduce an automated evaluation pipeline combining a VLM and an LLM to produce boolean pass/fail scores and natural language explanations. This setup enables joint evaluation of T2I and VLM models against human ground truth. The benchmark also supports difficulty scaling in prompts, revealing persistent weaknesses in tasks like counting and long-text generation. Overall, FineGRAIN offers a structured, interpretable, and extensible framework for evaluating generative models and highlights the limitations of current coarse-grained metrics.

**Dataset Code Accessibility:**

Yes

**Dataset Code Comments:**

The submission provides both the dataset and code in a publicly accessible and reproducible form. The dataset includes prompts, generated images from five T2I models, VLM/LLM annotations, and human-verified ground truth labels. The accompanying codebase supports generation, evaluation, and replication of the full pipeline, including the agentic scoring mechanism and templates for failure-mode-specific question generation. The authors also document the construction process and provide instructions for executing the benchmark. Overall, the dataset and code appear complete, well-organized, and sufficiently detailed to enable reproducibility.

**Ethical Considerations:**

No, there are no or only very minor ethics concerns

**Final Justification:**

The authors clarified my concern about synthetic image bias and circularity with a well-reasoned explanation on controlled failure modes and handcrafted prompts; the examples provided were helpful, and I update my score from 4 to 5.

**Limitations Weaknesses:**

1. The evaluation of VLMs is based entirely on synthetic images generated by T2I models. While this offers controlled prompt-image pairs, it also introduces the risk of circularity and bias—VLMs may learn to confirm prompt patterns rather than genuinely understand image content. The lack of evaluation on real-world imagery may also limit the generality of conclusions drawn about VLM capabilities.

2.  The instruction prompts used to query VLMs are generated from templates using an LLM, which may introduce abiases in how failure modes are elicited or judged. Since the quality of these prompts directly affects the reliability of VLM evaluation, it would be helpful for the authors to include representative examples of LLM-generated queries for each failure mode, either in the main paper or appendix.

3. The paper claims that prompt difficulty can be adjusted programmatically, but this is only demonstrated for two failure modes (counting and text generation). It remains unclear whether difficulty scaling is feasible for more abstract categories like spatial relations or cause-and-effect, and whether such prompts require manual design at each level. Clarifying how difficulty is defined and scaled across different failure types would strengthen the benchmark’s generality.

**Strengths Contributions:**

1. The paper is well-written and well-structured. The illustration of the FineGRAIN pipeline effectively conveys the overall benchmark design, clearly showing how the LLM and VLM components are integrated for automated failure detection.

2.  This paper addresses an interesting and meaningful question—how to evaluate vision-language models based on their ability to detect fine-grained visual failures in images. This perspective is valuable and novel, shifting the focus from high-level captioning tasks to detailed perceptual understanding and alignment.

3. This paper proposes a well-designed dataset composed of manually crafted prompts targeting 27 fine-grained failure modes, with images generated by five open-source T2I models and annotated with human-verified binary labels to support reliable and systematic evaluation. The dataset is a valuable contribution to the community.

---

> ### Author Rebuttal · Authors · 2025-07-31
>
> Thank you for taking the time to review our paper and recognizing that the dataset is a valuable contribution to the community in your review.
>
> 1. >The evaluation of VLMs is based entirely on synthetic images generated by T2I models. While this offers controlled prompt-image pairs, it also introduces the risk of circularity and bias—VLMs may learn to confirm prompt patterns rather than genuinely understand image content. The lack of evaluation on real-world imagery may also limit the generality of conclusions drawn about VLM capabilities.
>
> **The use of synthetic images allows for a controlled ground truth.** Real-world datasets lack prompt-image pairs where failure modes are known. An advantage of using synthetic data is that we are better at evaluating the VLM’s capabilities to act as a reward model to the T2I models than had the image been real. That said, for a more robust VLM evaluation, the inclusion of high-quality real-world data would benefit the evaluation.
>
> VLMs learning to confirm prompt patterns explicitly was a concern of ours when evaluating this pipeline. We noticed that for failure modes like counting, including the numbers from the original T2I prompt, often had a detrimental impact on the VLM's ability to correctly discern the failure mode. This was part of our reasoning as to **why each failure mode needs a handcrafted instruction prompt to be edited by an LLM** before being paired with the T2I image for the VLM to score.
>
> 2. >The instruction prompts used to query VLMs are generated from templates using an LLM, which may introduce abiases in how failure modes are elicited or judged. Since the quality of these prompts directly affects the reliability of VLM evaluation, it would be helpful for the authors to include representative examples of LLM-generated queries for each failure mode, either in the main paper or appendix.
>
> Yes we agree, it is currently in the github. While they couldn’t fit in the main paper they could be included in the appendix for the camera-ready version, one for each failure mode. Examples:
>
>
>     "prompt": "A table set with exactly seven blue plates and five red cups. Each item should be evenly spaced. Studio lighting, high resolution.",
>     "instruction_prompt": "Count how many plates are there? Count how many cups are there?"
>
>     "prompt": "A wooden apple with a smooth, shiny texture sitting next to a marble banana with a rough, veined surface. High detail.",
>     "instruction_prompt": "What is the texture of the apple? What is the texture of the banana?"
>
>     "prompt": "A landscape painting with digital mountains, a crayon-drawn river, and photorealistic trees.",
>     "instruction_prompt": "Instruction: What in the image is digital? What in the image is crayon-drawn? What in the image is photorealistic?"
>
>     "prompt": "A welcome sign at a historical landmark reading: \"Welcome to Our Nation's Heritage. As you walk through these halls, you are stepping into the pages of history. Each stone, each artifact tells a story of a time long past, a story that continues to shape our present and future.\"",
>     "instruction_prompt": "Perform OCR on the text in the image. Output only the text: \"[prompt]\""
>
> 3. >The paper claims that prompt difficulty can be adjusted programmatically, but this is only demonstrated for two failure modes (counting and text generation). It remains unclear whether difficulty scaling is feasible for more abstract categories like spatial relations or cause-and-effect, and whether such prompts require manual design at each level. Clarifying how difficulty is defined and scaled across different failure types would strengthen the benchmark’s generality.
>
> We focused on the ablations for counting and text rendering as they have a clear metric that is not simply based on human judgement or common subfailure modes within categories. Thus, those ablations are inherently relevant to those failure modes being numeric while other failure modes metrics are not. Scaling is possible for these more abstract categories however **to give a rating on what is hard for T2I models within a failure mode first requires their use to evaluate a diverse set of prompts** in a failure mode with our dataset. For example, an upside-down human is significantly more difficult for T2I models than and standing human or seemingly confusing or more abstract instructions. That said, it is hard to quantify how much more difficult into a level without then breaking the failure mode into another subcategory. During our refining process, we tried to combine these subcategories as they grew quite numerous. We do note that including a range of examples in the camera-ready version could be done for more of the failure modes.
>
>
>
> >Dataset Code
>
> Thank you for reviewing and verifying our dataset.

---

### Official Review · Reviewer_Ex74 · 2025-07-03

**Rating:** 5
**Confidence:** 3

**Summary:**

This paper addresses two key limitations: the inability of text-to-image (T2I) models to accurately follow prompt instructions, and the difficulty vision-language models (VLMs) face in understanding complex scenes. To address these challenges, the authors propose FineGRAIN, a novel evaluation framework that combines T2I generation and VLM-based assessment based on 27 fine-grained failure modes. The framework offers a structured pipeline and a refined dataset that enables joint analysis of both model types.

**Additional Feedback:**

1.	The paper does not clearly explain why Molmo 72B was selected as the VLM. Given its large size, it would have been valuable to discuss what trade-offs or limitations arise when using smaller models (e.g., LLaVA 7B, LLaVA 13B), especially in terms of T2I evaluation accuracy. This would provide helpful guidance on VLM selection.
2.	The key contribution of this paper lies in proposing a bidirectional evaluation framework for both T2I and VLM models. However, while a large VLM (Molmo 72B) was likely chosen to improve the reliability of T2I evaluation, a variety of T2I models were compared across scales. This raises the question of whether the framework is more heavily focused on evaluating T2I models. If the framework also aims to evaluate VLMs, it would have been better to include experiments that fix the T2I model and compare multiple VLMs.
3.	As mentioned earlier, the bidirectional nature of the framework introduces structural limitations. Nonetheless, it would be helpful if the paper better articulated the motivation for adopting a bidirectional design and what benefits it offers over a unidirectional approach.
4.	The paper lacks detailed information about the test environment (e.g., GPU specifications, memory requirements). Given the use of large-scale models, reporting computational cost or feasibility would improve the practical understanding of the framework.

**Dataset Code Accessibility:**

Yes

**Ethical Considerations:**

No, there are no or only very minor ethics concerns

**Final Justification:**

The additional evaluations of multiple VLMs significantly improved the robustness and reliability. The explanations regarding GPU resource requirements and practical considerations, along with clarification of the bidirectional framework’s motivations, effectively resolved the limitations I initially noted.

**Limitations Weaknesses:**

1.	The framework has reproducibility limitations, as performance comparisons are unreliable unless the VLM–T2I pair is fixed.
2.	While the paper claims to support VLM evaluation, only a single VLM is actually tested. Although this is mentioned as a limitation, evaluating at least two VLMs would have made the results more persuasive.
3.	The selected VLM, Molmo 72B, is a very large-scale model. It would have been helpful to include information about the minimum GPU requirements, inference time, or resource cost involved.

**Strengths Contributions:**

1.	The paper introduces the first unified framework that enables joint evaluation of both T2I models and VLMs. Unlike prior benchmarks that assessed these models independently, FineGRAIN integrates both into a single structured evaluation pipeline.
2.	Through prompt designs that deliberately induce failures and a taxonomy of 27 specific failure types, the framework allows detailed analysis of how and when models fail.
3.	Section 4 emphasizes that VLMs are not as reliable as humans in identifying failure modes. This raises a fundamental question about the validity of using VLMs as evaluators within the bidirectional framework.
4.	The systematic analysis of failure types helps readers clearly understand the limitations of current models.

---

> ### Author Rebuttal · Authors · 2025-07-31
>
> Thank you for the detailed review, which highlighted that our systematic analysis of failure types helps readers clearly understand the limitations of current models.
>
> 1. >  The framework has reproducibility limitations, as performance comparisons are unreliable unless the VLM–T2I pair is fixed.
>
> FineGRAIN supports both fixed and variable settings for reproducibility. For our final results, we used either the best-performing model or averaged across models. Our code release includes separate evaluation scripts for T2I-only and VLM-only settings. **Results in Table 5 were generated with fixed pairs for fair comparison.**
>
>
> 2. >  While the paper claims to support VLM evaluation, only a single VLM is actually tested. Although this is mentioned as a limitation, evaluating at least two VLMs would have made the results more persuasive.
>
> We completely agreed in this criticism at the time of the main paper submission. To address this, we have included a comparison in the submitted supplementary for **three state-of-the-art, open-source VLMs, Molmo-72B, InternVL3-78B, and Pixtral Large-124B, in Figure 4 of the appendix (p. 8)**. These models were chosen based on strong performance in our unit tests. We also provide unit tests with GPT-4 in Figure 2 (Appendix p. 2). **Performance was comparable across these VLMs, with Molmo slightly leading overall.** Thus, we used Molmo-72B, which was also the smallest of the three. This will be included in the camera-ready version.
>
> 3. > The selected VLM, Molmo 72B, is very large. It would have been helpful to include information about GPU requirements, inference time, or resource cost.
>
> To completely reproduce all VLM results will need **4-8×80GB GPUs** or an equivalent. Use of vllm or an equivalent can significantly affect the requirements of the implementation. While we choose open-source VLMs to keep the pipeline free, we do recognize not all researchers have access to the computational resources. That said, the major investment was in the iterations of crafting the dataset, a single inference pass for reproducibility or new models can the same large VLMs that are also available via APIs. The approximate VLM API cost for our dataset would be **about $8 USD**. We also provide an option for smaller models in our GitHub repo, though we do not recommend them due to severe performance drops in some failure modes.
>
> ### Additional Comments
>
> > The paper does not clearly explain why Molmo 72B was selected. It would have been helpful to discuss trade-offs or limitations of smaller models (e.g., LLaVA 7B, LLaVA 13B).
>
> Partially answered above. Our objective was to create the most useful benchmark for T2I models for the community, so we selected the **highest performing VLM to ensure reliable evaluation**. In our tests, larger VLMs consistently outperformed smaller ones. With smaller models having catastrophic failure for some failure modes.
>
>
> > The bidirectional nature of the framework introduces structural limitations. It would be helpful to better articulate the motivation and benefits of a bidirectional approach.
>
> We observed that our **automatic evaluation method introduces its own failure modes**, which need to be understood to gauge benchmark reliability. A bidirectional design allows us to expose and analyze these limitations. It further opens the possibility to use one model to reward the other where their failure modes differ. Furthermore, by jointly evaluating the failure modes of VLM’s and T2I models we can better understand the failure modes in Multimodal vision as a whole.
>
>
> > The paper lacks details about the test environment (e.g., GPU specs, memory).
>
> We used 8 80GB GPUs for the evaluations. API-based alternatives for the VLM's are feasible for users without such resources, costing around **$8 USD for the full dataset.**

---

> > ### Comment · Reviewer_Ex74 · 2025-08-02
> >
> > Thank you for clearly addressing my primary concerns through your detailed rebuttal. I have decided to increase my rating from 4 to 5

---

> > > ### Author Response · Authors · 2025-08-02
> > >
> > > Thank you for reviewing our rebuttal and updating your score. We are happy to hear your primary concerns were addressed.

---

### Decision · Program_Chairs · 2025-09-18

**Decision:**

Accept (spotlight)

**Comment:**

This paper proposes a novel framework for evaluating Text-to-Image models using Vision Language Models, thus building a unified framework to jointly identify the limitations of LLM and VLM automatically. By identifying the ability of vision language models to detect fine-grained visual failures in images, this work provides a useful tool for detailed perceptual understanding and alignment. All the reviewers appreciate the novelty of the paper and give a positive rating. The major initial weakness focuses on the evaluation side, for which additional evaluations during rebuttal further clarified the reviewer's concerns. The AC would like to recommend accepting it as an oral paper because it presents an interesting and worthwhile direction to explore.

===== FINAL UPDATE FROM DB Track PCs ====

The final decision for this paper has been taken by the program chairs after consultation with the SACs. All Senior Area Chairs have ranked papers according to the feedback from the AC during the review process. We decided to leave the original meta-review to reflect the opinion of the AC in light of the initial discussions with reviewers and SAC.